# STACKELBERG LEARNING FROM HUMAN FEEDBACK: PREFERENCE OPTIMIZATION AS A SEQUENTIAL GAME

**Barna Pásztor**[*]
ETH Zurich

**Thomas Kleine Buening**
ETH Zurich

**Andreas Krause**
ETH Zurich

## ABSTRACT

We introduce Stackelberg Learning from Human Feedback (SLHF), a new framework for preference optimization. SLHF frames the alignment problem as a sequential-move game between two policies: a Leader, which commits to an action, and a Follower, which responds conditionally on the Leader's action. This approach decomposes preference optimization into a refinement problem for the Follower and an optimization problem against an adversary for the Leader. Unlike Reinforcement Learning from Human Feedback (RLHF), which assigns scalar rewards to actions, or Nash Learning from Human Feedback (NLHF), which seeks a simultaneous-move equilibrium, SLHF leverages the asymmetry of sequential play to capture richer preference structures. The sequential design of SLHF naturally enables *inference-time refinement*, as the Follower learns to improve the Leader's actions, and these refinements can be leveraged through iterative sampling. We compare the solution concepts of SLHF, RLHF, and NLHF, and lay out key advantages in consistency, data sensitivity, and robustness to intransitive preferences. Experiments on large language models demonstrate that SLHF achieves strong alignment across diverse preference datasets, scales from 0.5B to 8B parameters, and yields inference-time refinements that transfer across model families without further fine-tuning.

## 1 INTRODUCTION

Reinforcement Learning from Human Feedback (RLHF) has emerged as the dominant paradigm for aligning Large Language Models (LLMs) with human preferences (Casper et al., 2023; Kaufmann et al., 2023). The standard pipeline involves two stages: first, a reward model is trained on a dataset of pairwise human comparisons, and second, a policy is optimized via reinforcement learning to maximize this reward (Christiano et al., 2017; Ouyang et al., 2022). Despite its empirical success, RLHF relies on a critical assumption that diverse human preferences can be faithfully represented by a single real-valued reward function. In practice, this assumption often fails as scalar reward models cannot capture *intransitive preference* structures. Even when preferences are transitive, widely used formulations such as the Bradley-Terry model (Bradley and Terry, 1952) can yield learned rewards that diverge from the underlying preferences (Bertrand et al., 2023).

A common alternative to reward models and the Bradley-Terry assumption are preference models which directly model pairwise preferences (Jiang et al., 2023). However, when preferences exhibit cycles, optimality becomes ill-defined because no single policy can dominate all others. Nash Learning from Human Feedback (NLHF) proposes the Nash Equilibrium (NE) as a solution to this problem by framing preference optimization as a two-player simultaneous-move game, where the Nash equilibrium (NE) corresponds to a typically stochastic policy whose actions are preferred to any other policy's actions on average (Munos et al., 2024).

We expand on this game-theoretic perspective and introduce Stackelberg Learning from Human Feedback (SLHF), which models alignment as a *sequential-move* game between a Leader and a Follower inspired by Stackelberg dynamics (Stackelberg, 1952). In SLHF, a Leader first commits to an action, and a Follower then responds conditional on the Leader's choice. This asymmetry yields two key advantages. First, the Follower solves a refinement problem rather than optimizing directly against a non-stationary opponent and unobserved actions. This leads to more stable learning and

---

[*]Correspondence to `barna.pasztor@ai.ethz.ch`

quicker adaptation to the changes in the Leader's policy. Consequently, this faster rate of learning yields a more stationary feedback to the Leader that can anticipate the Follower's refinement more accurately and choose actions that are robust to subsequent improvements.

Perhaps more importantly, SLHF provides a principled method for *inference-time refinement*: the ability to improve model outputs at inference-time via repeated sampling. This is particularly valuable when the target preferences change between training and inference-time. Most commonly, models are trained on preferences aggregated across diverse annotators that might induce intransitive preference cycles (Section 4). However, at inference-time, outputs ultimately have to align with an individual's taste. SLHF realizes this refinement through its two components: the Leader policy produces an initial response, and the Follower policy generates refined responses conditional on the previous output. Unlike sampling from a static distribution, this produces a sequence of outputs that can efficiently explore the preference space. Crucially, this allows for performance gains through inference-time computation alone, without any need for additional training or external feedback.

In summary, our contributions are as follows:

- We introduce Stackelberg Learning from Human Feedback (SLHF), a preference optimization framework that models alignment as a two-player sequential game. We formalize this game over a learned pairwise preference model and show that SLHF admits a unique Stackelberg equilibrium under standard regularity assumptions (Section 4).

- We propose STACKELBERGGDA, an algorithm that approximates the Stackelberg equilibrium via two-timescale gradient descent ascent. Our algorithm benefits from online RL optimization without the need of an explicit reward model or expensive inference with a mixture policy (Section 5).

- Our experimental results show that the Follower, conditioned on the Leader's output, consistently outperforms both RLHF and NLHF baselines, whereas the Leader performs similarly to the approximated Nash policy. Furthermore, we show that the Follower generalizes across models, improving outputs from independently trained policies without additional fine-tuning (Section 6).

## 2 RELATED WORK

**Reinforcement Learning from Human Feedback (RLHF).** RLHF optimizes policies using human preferences expressed through pairwise comparisons or rankings rather than explicit numeric rewards (Wirth et al., 2017; Kaufmann et al., 2023). The standard pipeline, introduced by Christiano et al. (2017), trains a reward model from human comparisons and then treats this model as a proxy reward for policy optimization, typically using PPO (Schulman et al., 2017). This framework has driven progress in text summarization (Stiennon et al., 2020), question answering (Nakano et al., 2021; Menick et al., 2022), and large language model fine-tuning (Ziegler et al., 2019; Bai et al., 2022; Glaese et al., 2022; Ouyang et al., 2022). Recent work integrates reward and policy updates into a bilevel optimization loop (Shen et al., 2024; Thoma et al., 2024; Makar-Limanov et al., 2024), but the reliance on a real-valued reward model remains. In particular, SGPO (Chu et al., 2025) considers a Stackelberg formulation between a policy and an adversarial preference distribution. In contrast, our work frames preference optimization as a sequential-move game between two policies and does not assume transitive preferences.

**Limitations of Reward Modeling.** Most RLHF implementations reduce preference learning to scalar reward estimation, typically based on the Bradley-Terry model (Bradley and Terry, 1952). While adequate for transitive, single-objective preferences, such models cannot represent intransitive structures and potentially misrank even transitive ones under model misspecification (Bertrand et al., 2023). Consequently, RLHF policies can be sensitive to the distribution of training comparisons (Munos et al., 2024) and prone to mode collapse under continued optimization (Xiao et al., 2024). Intransitive preference cycles have been observed not only in human feedback (Duan et al., 2017; Alós-Ferrer et al., 2022; Casper et al., 2023) but also in LLM-generated annotations (Dubois et al., 2024; Xu et al., 2025). Our approach sidesteps these issues by optimizing directly over pairwise preferences without imposing a scalar reward model.

**Preference Optimization.** To address the limitations of reward modeling in RLHF, IPO (Azar et al., 2023) extends Direct Preference Optimization (DPO) (Rafailov et al., 2023) by optimizing for the win rate against a reference policy. Nash Learning from Human Feedback (NLHF) casts the learning

problem as a two-player simultaneous-move game and introduces NASH-MD-PG and NASH-EMA-PG to approximate the Nash Equilibrium (NE) of a learned preference model via mirror descent (Munos et al., 2024). Subsequent work has extended this perspective, proposing various algorithms to optimize for (approximate) NE, including ONLINE-IPO (Calandriello et al., 2024), SPPO (Wu et al., 2024), SPO (Swamy et al., 2024), INPO (Zhang et al., 2025), DNO (Rosset et al., 2024), RSPO (Tang et al., 2025), NASH-RS (Liu et al., 2025) and MPO (Wang et al., 2025); sometimes with strong last-iterate convergence guarantees, e.g., EGPO (Zhou et al., 2025). Because simultaneous games are symmetric, these methods typically converge to mixed strategies unless one option is overwhelmingly preferred (Liu et al., 2025). In contrast, SLHF models alignment as a sequential Stackelberg game in which a Leader commits first and a Follower responds conditionally. This asymmetry yields a different solution concept that can admit deterministic equilibria in the non-regularized limit.

**Inference-Time Preference Improvement.** Improving the capabilities of LLMs through additional computation at inference time has recently received significant attention, especially in verifiable domains such as coding or mathematics (Welleck et al., 2024). Closest to our work are self-correction algorithms that aim to improve their responses without external feedback at test-time. A natural approach to self-correction is to provide instructions only without further training, which, however, can lead to performance degradation (Huang et al., 2024; Zheng et al., 2024; Tyen et al., 2024; Qu et al., 2024). Other work on training models for self-correction either assumes human or AI revisions (Saunders et al., 2022; Qu et al., 2024) or a reward function scoring responses (Welleck et al., 2023; Akyurek et al., 2023; Zhang et al., 2024; Kumar et al., 2025). Similarly to SLHF, Kumar et al. (2025) also propose to train an LLM in a sequential manner, however, assume a reward model and train in two-stages instead of a single loop. SLHF provides a unified alternative: its Leader-Follower structure naturally supports inference-time refinement through iterative sampling, enabling self-improvement on arbitrary preference signals without auxiliary reward models or multi-stage procedures.

## 3 PROBLEM STATEMENT

We consider a preference optimization problem over a finite set of contexts $\mathcal{X}$ and actions $\mathcal{Y}$. The contexts $x$ are drawn from a fixed and known distribution $\rho \in \Delta_{\mathcal{X}}$, where $\Delta_{\mathcal{X}}$ is the probability simplex over $\mathcal{X}$. A policy $\pi : \mathcal{X} \to \Delta_{\mathcal{Y}}$ maps each context $x \in \mathcal{X}$ to a discrete probability distribution $\pi(\cdot \mid x) \in \Delta_{\mathcal{Y}}$, where $\Delta_{\mathcal{Y}}$ is the probability simplex over $\mathcal{Y}$. We let $\Pi \coloneqq \{\pi \colon \mathcal{X} \to \Delta_{\mathcal{Y}}\}$ denote the set of all policies. In the language modeling setting, $\mathcal{X}$ typically models the set of prompts, $\mathcal{Y}$ the candidate responses, and $\pi$ is the LLM that defines a conditional distribution over responses given prompts.

Let the *preference function* $p(y \succ y' \mid x)$ define the probability that $y$ is preferred over $y'$ given $x$. We adopt the convention of writing $y \succ_x y'$ when $p(y \succ y' \mid x) > {}^1\!/_2$. Slightly overloading notation, the preference between two policies $\pi$ and $\pi'$ given context $x$ is defined as

$$p(\pi \succ \pi' \mid x) \coloneqq \mathbb{E}_{y \sim \pi(\cdot|x), y' \sim \pi'(\cdot|x)}\big[p(y \succ y' \mid x)\big]. \tag{1}$$

There are two common approaches to implementing the preference function $p$ in practice. Let $\mathcal{D} = \big\{(x_i, y_i^w, y_i^l)\big\}_{i=1}^N$ be a preference dataset, where $y_i^w$ and $y_i^l$ denote the chosen and rejected actions in a pairwise comparison, respectively. One approach is to frame this as a binary classification problem on $\mathcal{D}$ and train a parametrized model to estimate $p$ (Jiang et al., 2023). Alternatively, in the language modeling setting, one could directly employ trained models to provide feedback by following instructions without additional training. This method is often referred to as LLM-as-a-judge (Gu et al., 2025).

The core objective of preference optimization is to identify a policy that consistently generates optimal or highly-preferred responses. The notion of an "optimal" policy is straightforward when preferences are transitive. In such a case, for a given context $x \in \mathcal{X}$, there exists an action $y_x^\star \in \mathcal{Y}$ such that $y_x^\star \succ_x y$ for all $y \in \mathcal{Y}$. This action is known as a *Condorcet winner* for $x$ and represents the top element of the induced total order. If every context $x \in \mathcal{X}$ admits a Condorcet winner, the optimal policy is simply $\pi^\star(x) = y_x^\star$. However, real preference data often contains cycles or other intransitivities, so a Condorcet winner may not exist and policy optimality becomes ill-defined. To cope with such ambiguity, prior work adopts different solution concepts, the two most common of which we briefly review below.

### 3.1 BACKGROUND ON EXISTING SOLUTION CONCEPTS AND APPROACHES

**Reinforcement Learning from Human Feedback (RLHF).** RLHF as proposed by Christiano et al. (2017) and adapted to language modeling by Ziegler et al. (2019) splits preference optimization into

two steps. First, it assumes that the preference function $p$ follows the Bradley-Terry model (Bradley and Terry, 1952) so that

$$p(y \succ y' \mid x) = \sigma(r(x, y) - r(x, y')), \quad (2)$$

where $\sigma(x) = \frac{1}{1+\exp(-x)}$ is the sigmoid function and $r : \mathcal{X} \times \mathcal{Y} \to \mathbb{R}$ is a real-valued reward function. The reward function $r$ is unknown so that an estimator $\hat{r}$ is used that maximizes the log-likelihood of the dataset $\mathcal{D}$. In a second step, the policy $\pi^\star$ is chosen to maximize the expected reward with respect to $\hat{r}$ regularized by the Kullback-Leibler (KL) divergence against a fixed reference policy $\pi^{\text{ref}} \in \Pi$:

$$\pi^\star \in \arg\max_{\pi \in \Pi} \mathbb{E}_{x \sim \rho} \Big[ \mathbb{E}_{y \sim \pi(\cdot|x)} \big[ \hat{r}(x, y) \big] - \tau \mathrm{KL}_x(\pi \,\|\, \pi^{\text{ref}}) \Big]. \quad (3)$$

Here, $\tau \geq 0$ and $\mathrm{KL}_x(\pi \,\|\, \pi^{\text{ref}})$ is computed between $\pi(\cdot \mid x)$ and $\pi^{\text{ref}}(\cdot \mid x)$. Under the Bradley-Terry assumption, Equation (3) admits a unique closed-form solution (Rafailov et al., 2023). However, additive score models like Bradley-Terry are provably limited in expressing cyclic or intransitive preference structures, which have been empirically observed in both strategic games (Bertrand et al., 2023) and human preference data (Duan et al., 2017; Alós-Ferrer et al., 2022; Casper et al., 2023). Consequently, the optimal policy $\pi^\star$ depends critically on the data distribution in the training set $\mathcal{D}$, especially its sampling biases (Munos et al., 2024), which we elaborate more on in Section 4.1.

**Nash Learning from Human Feedback (NLHF).** NLHF avoids explicit reward modeling by framing preference optimization as a two-player simultaneous game between two policies $\pi, \pi' \in \Pi$ (Munos et al., 2024). The optimization problem is given by:

$$\max_{\pi \in \Pi} \min_{\pi' \in \Pi} \mathbb{E}_{x \sim \rho} \Big[ p(\pi \succ \pi' \mid x) - \tau \mathrm{KL}_x(\pi \,\|\, \pi^{\text{ref}}) + \tau \mathrm{KL}_x(\pi' \,\|\, \pi^{\text{ref}}) \Big]. \quad (4)$$

The solution to Equation (4) is a *Nash equilibrium* $(\pi^\star, \pi'^\star)$, where neither side can be improved unilaterally. The existence and uniqueness of this equilibrium follows from the concave-convex nature of the objective (Munos et al., 2024). NLHF can incorporate online feedback and makes no structural assumptions on preferences, but when no action is majority-preferred the equilibrium necessarily involves mixed strategies (Liu et al., 2025), even in the absence of KL regularization when $\tau = 0$. This inherent stochasticity can be undesirable in applications where consistency and reliability are critical.

## 4 STACKELBERG LEARNING FROM HUMAN FEEDBACK (SLHF)

We now present Stackelberg Learning from Human Feedback (SLHF), a novel perspective on the preference optimization problem. Inspired by Stackelberg games (Stackelberg, 1952), we cast preference optimization as a *sequential-move* game between two players: the *Leader* and the *Follower*. Given a context $x$, the Leader first chooses its action $y \sim \pi(\cdot \mid x)$. The Follower then observes both the context $x$ and the Leader's realized action $y$ and responds with $y' \sim \omega(\cdot \mid x, y)$. The Follower's policy $\omega$ is chosen from the set $\Omega = \{\omega : \mathcal{X} \times \mathcal{Y} \to \Delta_{\mathcal{Y}}\}$ which allows conditioning on both the context and the Leader's action. Formally, given reference policies $\pi^{\text{ref}} \in \Pi$ and $\omega^{\text{ref}} \in \Omega$, the optimization problem is defined as follows:

$$\max_{\pi \in \Pi} \min_{\omega \in \Omega} \mathbb{E}_{x \sim \rho} \Big[ \mathbb{E}_{y \sim \pi(\cdot|x)} \big[ \mathbb{E}_{y' \sim \omega(\cdot|x,y)} [p(y \succ y' \mid x)] + \tau^F \mathrm{KL}_{x,y}(\omega \,\|\, \omega^{\text{ref}}) \big] - \tau^L \mathrm{KL}_x(\pi \,\|\, \pi^{\text{ref}}) \Big] \quad (5)$$

where $\tau^L, \tau^F \geq 0$ are player-specific regularization coefficients. We let $f(\pi, \omega)$ denote the objective of Equation (5), which, in the absence of regularization, defines a sequential-move constant-sum game.

SLHF decomposes the preference optimization into two complementary roles, setting it apart from single-policy methods like RLHF and NLHF. The Follower leverages its informational advantage of observing the Leader's committed action. This simplifies its task to learning a specialized refinement policy that finds the best response to a known output, rather than optimizing against a non-stationary opponent. The Leader, anticipating this refinement, learns to produce initial actions that remain strong even after the Follower's refinement. In Section 4.1, we illustrate that when preferences form a cycle and no Condorcet winner exists, the Leader selects the least exploitable action, while the Follower traverses the preference cycle, covering all plausibly optimal actions with minimal samples.

The formulation in (5) differs from standard Stackelberg settings (Conitzer and Sandholm, 2006), where the Follower gets to condition on the Leader's policy $\pi$ only, not on the realized action $y$.

Table 1: Transitive individual annotator preferences over three options $\{A, B, C\}$.

| Type | Preference Relationship | Proportion |
|------|-------------------------|------------|
| $a_1$ | $A \succ B \succ C$ | $\alpha_1$ |
| $a_2$ | $B \succ C \succ A$ | $\alpha_2$ |
| $a_3$ | $C \succ A \succ B$ | $\alpha_3$ |

Table 2: The preference function $p$ induced by the population in Table 1.

| | $A$ | $B$ | $C$ |
|---|-----|-----|-----|
| $A$ | 0.5 | $1 - \alpha_2$ | $\alpha_1$ |
| $B$ | $\alpha_2$ | 0.5 | $1 - \alpha_3$ |
| $C$ | $1 - \alpha_1$ | $\alpha_3$ | 0.5 |

Allowing the Follower to observe $y$ provides strictly more information whenever $\pi$ is stochastic, yielding a simpler and stationary best response problem. In this setting, the Leader gains no advantage from randomizing, i.e., playing a stochastic policy.

In line with previous results that the RLHF problem (3) admits a closed-form solution, we show that there exists a unique solution to the SLHF problem (5). The proof is deferred to the Section A.1.

**Proposition 1.** *Let $\tau^L, \tau^F > 0$ and suppose that $\pi^{\mathrm{ref}}(y \mid x) > 0$ for all $(x, y) \in \mathcal{X} \times \mathcal{Y}$. For any preference function $p(y \succ y' \mid x)$ there exists a unique solution $(\pi^\star, \omega^\star)$ to the preference optimization problem in Equation (5).*

The solution $(\pi^\star, \omega^\star)$ is called a *Stackelberg equilibrium*. It is folklore in the algorithmic game theory literature that there exists a deterministic Stackelberg equilibrium when the Leader's realized action is observed by a best responding Follower, as there always exists a deterministic best response for the Follower. Thus, there is no point in randomizing for the Leader. This stands in contrast to the NE, which is in general stochastic. For completeness, we provide a proof in Appendix A.2.

**Remark 2.** *For any preference function $p(y \succ y' \mid x)$, the SLHF optimization problem (5) has a deterministic solution $(\pi^\star, \omega^\star)$ whenever $\tau^L = \tau^F = 0$. Note that this solution may not necessarily be unique due to the lack of regularization.*

### 4.1 COMPARISON OF SOLUTION CONCEPTS

Before describing how to approximate the Stackelberg equilibrium, we first contrast RLHF, NLHF, and SLHF in the Condorcet paradox (de Caritat Mis et al., 1785) described below. Consider a setting with a single context $|\mathcal{X}| = 1$ and three candidate actions $\mathcal{Y} = \{A, B, C\}$. Let the preference function $p$ be given by the aggregate over the population of annotators, $\mathcal{A} = \{a_1, a_2, a_3\}$, defined in Table 1. Each type of annotator has a strict preference ranking over $\mathcal{Y}$ and we aggregate their preferences as

$$p(y \succ y') = \sum_{i=1}^{3} \alpha_i \mathbf{1}\{y \succ_{a_i} y'\}, \tag{6}$$

where $\mathbf{1}\{y \succ_{a_i} y'\}$ is 1 if $y$ is preferred over $y'$ by the annotator type $a_i$ and 0 otherwise. Table 2 shows the aggregated preferences of the whole population. For example, $p(A \succ B)$ is the probability that a randomly chosen annotator prefers $A$ to $B$. This is true for annotator types $a_1$ and $a_3$ (from Table 1), who make up a proportion $\alpha_1 + \alpha_3$ of the population. Since $\alpha_1 + \alpha_2 + \alpha_3 = 1$, this is equivalent to $1 - \alpha_2$, as shown in Table 2. A common example is to choose $\alpha_1 = \alpha_2 = \alpha_3 = 1/3$, which leads to a cyclic relationship between three actions where $A \succ B \succ C$ but $C \succ A$. Hence, there exists no Condorcet winner in this case. More generally, the interesting case is given by $\alpha_1, \alpha_2, \alpha_3 < 1/2$, and this example is often referred to as the *Condorcet paradox*, because the annotators individually have transitive preferences (Table 1), but their aggregated preferences form a cycle (Table 2). For ease of presentation, we consider a non-regularized problem in the rest of this section so that $\tau = \tau^L = \tau^F = 0$.

**RLHF Solution.** Our first observation is that the estimated reward function $\hat{r} : \mathcal{X} \times \mathcal{Y} \to \mathbb{R}$ depends on the sampling distribution of the dataset $\mathcal{D}$ used to estimate $\hat{r}$. Suppose $\mathcal{D}$ contains only comparisons $\{A, B\}$ and $\{B, C\}$, but not $\{A, C\}$. Because $A \succ B$ and $B \succ C$ for all annotators, maximum-likelihood estimation, which fits a single underlying transitive reward function, yields $\hat{r}(A) > \hat{r}(B) > \hat{r}(C)$, so the optimal policy is $\pi^\star(A) = 1$. However, different sampling patterns (e.g., omitting $\{A, B\}$) can instead favor $B$ or $C$. This illustrates a key limitation of RLHF, as its solutions are sensitive to the specific comparisons present in $\mathcal{D}$.

---

**Algorithm 1** STACKELBERGGDA

---

1: **procedure** STACKELBERGGDA($\mathcal{X}, \mathcal{Y}, \eta^L, \eta^F$)
2:     Initialize the Leader and Follower policies $\pi_1$ and $\omega_1$
3:     **for** $i = 1, 2, \ldots$ **do**
4:         Update Leader's policy: $\pi_{i+1} = \pi_i + \eta^L \nabla_\pi f(\pi_i, \omega_i)$
5:         Update Follower's policy: $\omega_{i+1} = \omega_i - \eta^F \nabla_\omega f(\pi_i, \omega_i)$
6:         Project $\pi_{i+1}$ and $\omega_{i+1}$ to their respective probability simplices
7:     **end for**
8: **end procedure**

---

**Nash Equilibrium.** The NE of the matrix game defined in Table 2 is given by $\pi^\star(A) = 1 - 2\alpha_3$, $\pi^\star(B) = 1 - 2\alpha_1$, $\pi^\star(C) = 1 - 2\alpha_2$. In the special case of $\alpha_1 = \alpha_2 = \alpha_3 = 1/3$, the NE is uniform over $\mathcal{Y}$, i.e., it has the highest possible entropy. Unlike RLHF, this solution is dataset-independent, but it produces a fully stochastic policy that may be undesirable in applications requiring decisive outputs.

**Stackelberg Equilibrium.** In SLHF, the players' sequential roles resolve the cycle. The Follower's optimal strategy is straightforward as for any action $y$ presented by the Leader, it plays the best response $y'$ that beats it (i.e., $C$ if $y = A$, etc.). The Leader, anticipating this deterministic best response, chooses an initial robust action. This leads to the following equilibrium policies:

$$\omega^\star(\cdot \mid y) = \begin{cases} C & \text{if } y = A \text{ w.p. } 1 \\ A & \text{if } y = B \text{ w.p. } 1 \\ B & \text{if } y = C \text{ w.p. } 1 \end{cases} \qquad \pi^\star(\cdot) = \begin{cases} A & \text{if } \alpha_1 > \max\{\alpha_2, \alpha_3\} \text{ w.p. } 1 \\ B & \text{if } \alpha_2 > \max\{\alpha_1, \alpha_3\} \text{ w.p. } 1 \\ C & \text{if } \alpha_3 > \max\{\alpha_1, \alpha_2\} \text{ w.p. } 1 \end{cases}.$$

When $\alpha_1 = \alpha_2 = \alpha_3 = 1/3$, the Leader is indifferent, and any distribution over $A, B, C$ (including the uniform NE) is a valid Stackelberg equilibrium. Unlike RLHF, this solution requires no offline dataset, and unlike NLHF, it admits a deterministic Leader and Follower policy when one type dominates.

**Inference-Time Refinement.** Motivated by applications such as text summarization, open-ended generation, and audio-visual content creation, where users can reject outputs and request new samples, we introduce the notion of *inference-time refinement* for preference optimization. At inference-time, a single user interacts with the model and may resample actions until receiving one that matches their preference, analogous to the $pass@k$ metric in verifiable domains. This is non-trivial because models are usually trained to reflect *population preferences*, yet deployment requires adaptation to an *individual user*.

Consider the symmetric case $\alpha_1 = \alpha_2 = \alpha_3 = 1/3$, and without loss of generality, let the user be of type $a_1$ with ranking $A \succ B \succ C$. RLHF may return $A$, but depending on $\mathcal{D}$ it could also output $B$ or $C$, and repeated sampling offers no recourse. The NLHF solution is uniform over $A, B, C$, so the probability of sampling $A$ in a single draw is $1/3$. By sampling $N$ times, the probability of observing at least one $A$ is $1 - (2/3)^N$, i.e., 56% for $N = 2$ and 70% for $N = 3$. The SLHF solution starts similarly: the first action is sampled from the Leader's possibly uniform policy but subsequent actions are drawn from the Follower's policy, i.e., $y_i \sim \omega^\star(\cdot \mid x, y_{i-1})$ for $i \geq 2$. Following this structure, the probability of sampling $A$ within $N = 2$ steps increases to 67%, and for $N = 3$, the entire preference cycle is traversed regardless of the Leader's initial choice. Note that the SLHF solution supports this refinement procedure without the need of additional training required at inference-time.

| User: `<user_prompt>` |
|---|
| Assistant: |

(a) Prompt received as the Leader agent

| User: `<user_prompt>` |
|---|
| Assistant: `<leader_response>` |
| User: Improve the previous answer! |
| Assistant: |

(b) Prompt received as the Follower agent

Figure 1: Prompt templates used to train a single-model for both Leader and Follower completions.

Table 3: Pairwise preference comparisons between the responses of QWEN2.5-0.5B, RLOO, NASH-MD-PG, and STACKELBERGGDA algorithms. Each cell represents the preference model's average score for the row algorithm over the column algorithm.

| | QWEN2.5-0.5B | RLOO | NASH-MD-PG | STACKELBERGGDA | |
| --- | --- | --- | --- | --- | --- |
| | | | | LEADER | FOLLOWER |
| QWEN2.5-0.5B | 0.000 | 0.407 | 0.279 | 0.266 | 0.200 |
| RLOO | 0.593 | 0.000 | 0.393 | 0.387 | 0.344 |
| NASH-MD-PG | **0.721** | 0.607 | 0.000 | 0.497 | 0.406 |
| STACKELBERGGDA-LEADER | **0.734** | 0.613 | **0.503** | 0.000 | 0.395 |
| STACKELBERGGDA-FOLLOWER | **0.800** | **0.656** | **0.594** | **0.605** | 0.000 |

## 5 STACKELBERG GRADIENT DESCENT ASCENT (STACKELBERGGDA)

We now introduce STACKELBERGGDA, a two-timescale Gradient Descent-Ascent (GDA) algorithm designed for the sequential-move preference optimization problem in Section 4. STACKELBERGGDA performs simultaneously gradient ascent and descent update steps on the Leader and Follower policies, $\pi$ and $\omega$, with step size $\eta^L$ and $\eta^F$, respectively, to find the $\max\min$ solution to $f(\pi, \omega)$ defined in Equation (5). It is a two-timescale algorithm as we choose $\eta^F > \eta^L$ resulting in $\omega$ adapting faster than $\pi$. We denote the two-timescale coefficient as $\kappa = \eta^F/\eta^L$. After each update, both policies are projected back onto their respective probability simplices to ensure feasibility.

The function $f(\pi, \omega)$ is concave in $\pi$ and convex in $\omega$.[1] While standard gradient descent-ascent with equal learning rates has ergodic convergence guarantees in this setting (Korpelevich, 1976; Chen and Rockafellar, 1997; Nemirovski, 2004; Auslender and Teboulle, 2009; Nedić and Ozdaglar, 2009), we instead adopt a two-timescale variant. This choice is motivated by its stronger convergence guarantees in more general nonconvex-concave regimes (Lin et al., 2025), as well as its empirical success in both Actor-Critic methods (Prasad et al., 2015) and the training of Generative Adversarial Networks (Heusel et al., 2017). This becomes especially valuable for the practical implementation of STACKELBERGGDA for large state and action spaces and parameterized policies below.

**Scalable Implementation of STACKELBERGGDA for LLM Fine-Tuning.** Direct optimization over the full policy spaces $\Pi$ and $\Omega$ is infeasible when $\mathcal{X}$ and $\mathcal{Y}$ are large, as in LLM fine-tuning. To address this challenge, we parametrize $\pi$ and $\omega$ and estimate gradients from batches. Crucially for LLM fine-tuning, the Leader and the Follower can share the same parametrization by using the prompt template shown in Figure 1, which allows us to reduce the memory requirements. Additionally, framing the Leader and the Follower policies as multi-turn dialogues enables us to use any policy trained for multi-turn conversations as both $\pi^{\text{ref}}$ and $\omega^{\text{ref}}$. All implementation details and pseudocode are provided in Section B.

## 6 EXPERIMENTS

We conduct a series of experiments to validate the Stackelberg formulation and the efficacy of STACKELBERGGDA. Our evaluation is designed to answer three primary questions:

1) How does STACKELBERGGDA compare against established RLHF and NLHF baselines in a controlled preference optimization task?

2) Can the Leader-Follower structure of SLHF enable effective inference-time refinement, and does this capability generalize to improving outputs from other models?

3) Does the approach scale effectively to the large-scale, general-purpose fine-tuning of LLMs?

Section 6.1 addresses the first two questions by aligning models on a dataset with diverse human preference signals. In the appendix, we also provide further results on iterative improvements with increased inference-time computation (Section D.2), ablations of the hyperparameter $\kappa$ (Section D.3),

---

[1]This follows from Munos et al. (2024). For completeness, we provide a formal proof in Section A.3, and discuss the convergence behavior and limitations of STACKELBERGGDA in Section A.4.

Table 4: Test-time improvement using different models for the initial response (Leader) and the improvement (Follower). Each cell represents the preference model's average score for the Follower's responses over the Leader's responses.

| | | Leader | | | |
|---|---|---|---|---|---|
| | | QWEN2.5-0.5B | RLOO | NASH-MD-PG | STACKELBERGGDA |
| **Follower** | QWEN2.5-0.5B | 0.549 | 0.443 | 0.363 | 0.362 |
| | RLOO | 0.534 | 0.403 | 0.369 | 0.360 |
| | NASH-MD-PG | 0.708 | 0.600 | 0.493 | 0.476 |
| | STACKELBERGGDA | **0.803** | **0.665** | **0.600** | **0.606** |

and additional scaling results (Section D.4). Section 6.2 then tackles the third question by applying STACKELBERGGDA within a large-scale, open-source post-training pipeline.

## 6.1 EMPIRICAL COMPARISON OF SOLUTION CONCEPTS

**Dataset.** We use the HELPSTEER2 dataset (Wang et al., 2024), which contains 11,826 human-annotated single-turn dialogues, to estimate the preference function $p$ and its prompts during the training loops. We choose this dataset due to its high-quality human annotations along five attributes (helpfulness, correctness, coherence, complexity, and verbosity) that allows us to estimate a diverse preference profile. Further details on the preference model specification and the resulting intransitivity are provided in Section D.1.

**Compared Methods.** We compare STACKELBERGGDA with RLOO (Ahmadian et al., 2024) and NASH-MD-PG (Munos et al., 2024) which represent the RLHF and NLHF frameworks, respectively. We use these baselines because they are well-established and come with robust, well-tested open-source implementations. This ensures that our comparison reflects differences between the frameworks rather than implementation details. All models are fine-tuned from the QWEN2.5-0.5B[2] model and run for 1,000 gradient steps with a batch size of $B = 32$. We sweep learning rates $\eta \in \{1e-6, 5e-6, 1e-5\}$ and KL penalties $\tau \in \{0.001, 0.01, 0.1\}$ for all algorithms. For NASH-MD-PG, we additionally vary the mixture parameter $\beta \in \{0, 0.25, 0.5, 0.75, 1\}$, and for STACKELBERGGDA, the two-timescale coefficient $\kappa \in \{1, 5, 10\}$. Models are selected by average preference rate over QWEN2.5-0.5B yielding best setting $\eta = 1e^{-5}$ and $\tau = 0.001$, with $\beta = 0.75$ for NASH-MD-PG and $\kappa = 5$ for STACKELBERGGDA. All implementations use the `Transformers` (Wolf et al., 2020) and `TRL` (von Werra et al., 2020) libraries, with the AdamW optimizer (Loshchilov and Hutter, 2019).

### 6.1.1 ROUND-ROBIN TOURNAMENT

Table 3 reports pairwise preference scores between the initial QWEN2.5-0.5B and the three fine-tuned models. The first responses of STACKELBERGGDA-LEADER and NASH-MD-PG achieve roughly 73% preference over QWEN2.5-0.5B and 61% over RLOO, while tying at 50% when compared to each other. This outcome aligns with settings where multiple high-quality responses exist and the Stackelberg and Nash equilibria coincide (Section 4.1).

Crucially, applying the FOLLOWER of STACKELBERGGDA to improve its own initial responses yields a marked performance gain. It achieves 80% preference over QWEN2.5-0.5B, 66% over RLOO, 60% over NASH-MD-PG, and even outperforms the responses it was conditioned on in 60.5% of comparisons. Thus, a two-turn inference procedure provides substantial gains at the cost of a single additional generation.

### 6.1.2 INFERENCE-TIME REFINEMENT

We further evaluate each model's ability to act as a Follower, refining outputs from other models. Although only STACKELBERGGDA is explicitly trained for this task (and only to best

---

[2]https://huggingface.co/unsloth/Qwen2.5-0.5B-Instruct

Table 5: Benchmark evaluation results for AlpacaEval 2.0 and IFEval. Length-controlled (LC) winrate for AlpacaEval 2.0 alleviate the length bias of the GPT-4 judge.

| Model | AlpacaEval 2.0 | | IFEval |
|---|---|---|---|
| | LC Winrate | Winrate | Prompt Loose Acc. |
| GEMMA-2-9B-IT | **48.18** | 36.99 | 71.53 |
| STACKELBERGGDA-FOLLOWER | **44.57** | 34.47 | 61.92 |
| STACKELBERGGDA-LEADER | 35.04 | 25.59 | 71.71 |
| LLAMA-3.1-TULU-3-8B-DPO | 33.37 | **40.15** | **75.23** |
| QWEN2.5-7B-INSTRUCT | 29.52 | 29.91 | **73.01** |
| LLAMA-3.1-8B-INSTRUCT | 24.66 | 26.69 | 71.72 |
| LLAMA-3.1-TULU-3-8B-SFT | 8.83 | 14.26 | 67.46 |

respond to itself), we apply the same refinement procedure to all models to test their ability to generalize. Specifically, for every pair of Leader and Follower models selected from QWEN2.5-0.5B, RLOO, NASH-MD-PG, STACKELBERGGDA, we first generate a response with the selected Leader and then apply the Follower prompting template (Figure 1(b)) to produce a potentially improved response. We refer to these as the Leader and Follower outputs, respectively. Exhaustively evaluating all Leader-Follower pairs allows us to measure each model's capacity for inference-time refinement under diverse initial conditions. Table 4 reports the resulting preference scores, which indicate how often the Follower output is preferred over the Leader's generation.

STACKELBERGGDA consistently improves across all Leader models; most notably over QWEN2.5-0.5B and RLOO, while achieving gains of up to $60\%$ even when refining outputs from NASH-MD-PG or itself. In contrast, QWEN2.5-0.5B and RLOO only improve upon responses from QWEN2.5-0.5B and often degrade the quality of outputs from other Leaders. NASH-MD-PG can enhance responses from QWEN2.5-0.5B and RLOO, but its $70\%$ preference score over QWEN2.5-0.5B still falls short of its own $73\%$ self-improvement rate reported in Table 3. These findings extend prior work on verifiable domains (Huang et al., 2024; Zheng et al., 2024; Tyen et al., 2024; Qu et al., 2024) by showing that explicitly training to improve given outputs is crucial and mere instruction prompting is insufficient to reliably enhance responses with respect to human preferences.

## 6.2 GENERAL PURPOSE FINE-TUNING

To demonstrate the efficacy of STACKELBERGGDA for large-scale fine-tuning in general chat applications, we train the Tulu3 Supervised Fine-Tuned (SFT) checkpoint[3], denoted LLAMA-3.1-TULU-3-8B-SFT, with STACKELBERGGDA using the prompts from the preference dataset released with the model[4] (Lambert et al., 2025). We use the SKYWORK-CRITIC-LLAMA-3.1-70B model to provide pairwise feedback during training (Shiwen et al., 2024), and evaluate the resulting models on AlpacaEval 2.0, a benchmark shown to approximate human judgments in pairwise comparisons (Dubois et al., 2024), and IFEval, which measures verifiable instruction-following ability (Zhou et al., 2023).

As shown in Table 5, both STACKELBERGGDA-LEADER and STACKELBERGGDA-FOLLOWER substantially improve the AlpacaEval 2.0 win rates of the base model and outperform other models of similar scale, except for GEMMA-2-9B-IT. Notably, LLAMA-3.1-TULU-3-8B-DPO was trained from the same base model and prompt dataset but used Direct Preference Optimization (Rafailov et al., 2023) with completions and feedback from frontier models. This yields higher IFEval scores, reflecting stronger verifiable instruction following, whereas STACKELBERGGDA achieves substantially higher AlpacaEval 2.0 win rates, a closer proxy for human preferences and the main focus of this work.

Interestingly, during training we observed that the base model exhibits a notable drop in IFEval accuracy when used as the Follower policy, from $67.46\%$ to $59.89\%$, which may result from the increased context length. Nevertheless, instruction-following performance can be recovered and further improved through reinforcement learning with a verifiable reward. We view the integration of verifiable and comparative feedback within the SLHF framework as a promising direction for future research.

---

[3]https://huggingface.co/allenai/Llama-3.1-Tulu-3-8B-SFT
[4]https://huggingface.co/datasets/allenai/llama-3.1-tulu-3-8b-preference-mixture

## 7 CONCLUSION

We introduced Stackelberg Learning from Human Feedback (SLHF), a two-player sequential-move framework that directly optimizes pairwise preference signals without requiring real-valued reward models. We proposed STACKELBERGGDA to efficiently approximate the unique Stackelberg equilibrium and scale to challenging tasks such as aligning LLMs with human preferences. Empirically, STACKELBERGGDA's Leader policy matches or exceeds standard baselines while the Follower policy consistently improves outputs at inference time, even when paired with models it was not trained with.

**Limitations.** Similarly to NLHF, a key limitation of our approach is its reliance on a well-specified and representative pairwise preference function, which can be challenging to obtain in open-ended or under-specified domains. Moreover, although the sequential formulation enables inference-time refinement through conditional generation, it currently operates without real-time user interaction. Future work could integrate active preference elicitation and personalized refinement, allowing SLHF to adapt dynamically to individual user preferences at test time. Finally, STACKELBERGGDA currently has ergodic but not last-iterate guarantees; developing SLHF algorithms with last-iterate convergence (e.g., via extragradient/optimistic or mirror-prox) is an open direction.

## ACKNOWLEDGEMENTS

This work was primarily supported by the ETH AI Center through an ETH AI Center Doctoral Fellowship to Barna Pásztor and an ETH AI Center Postdoctoral Fellowship to Thomas Kleine Buening. Thomas Kleine Buening was additionally supported by the EPSRC Prosperity Partnership FAIR (grant number EP/V056883/1). We thank Marian Schneider for his support in the experiments' technical setup.

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

## CONTENTS OF APPENDIX

## A PROOFS

### A.1 PROOF OF THEOREM 1

*Proof.* First, assume that the Leader's policy $\pi$ is fixed and consider the Follower's optimization problem

$$\min_{\omega} \mathbb{E}_{x \sim \rho, y \sim \pi(\cdot|x)} \left[ \mathbb{E}_{y' \sim \omega(\cdot|x,y)}[p(y \succ y' \mid x)] + \tau^F \mathrm{KL}_{x,y}(\omega \| \omega^{\mathrm{ref}}) \right]. \tag{7}$$

The optimization problem in (7) is equivalent to Equation (3) for the reward function $r(\tilde{x}, y') := p(y \succ y' \mid x)$ with contexts $\tilde{x} = (x, y)$ and context distribution $\tilde{x} \sim \rho \otimes \pi$. As a result, Equation (7) has a unique closed-form solution (Geist et al., 2019; Rafailov et al., 2023; Azar et al., 2023) given by

$$\omega^{\star}(y' \mid x, y) = \frac{1}{Z(x, y)} \omega^{\mathrm{ref}}(y' \mid x, y) \exp\left(\tfrac{1}{\tau^F} p(y' \succ y \mid x)\right)$$

where $Z(x, y) = \sum_{y' \in \mathcal{Y}} \omega^{\mathrm{ref}}(y' \mid x, y) \exp\left(\tfrac{1}{\tau^F} p(y' \succ y \mid x)\right)$ is a partition factor that depends only on $(x, y)$ and $\pi^{\mathrm{ref}}$. Hence, $\omega^{\star}$ can be expressed as a function of $(x, y)$ and $\omega^{\mathrm{ref}}$ without explicit dependence on $\pi$.

Now, define the following reward function for the Leader's optimization problem

$$r(x, y) := \mathbb{E}_{y' \sim \omega^{\star}(\cdot|x,y)}[p(y \succ y' \mid x)]. \tag{8}$$

Note that $\omega^{\star}$ is unique so that $r(x, y)$ is a scalar. We can now restate Equation (5) for the Leader's optimization problem as

$$\max_{\pi} \mathbb{E}_{x \sim \rho} \left[ \mathbb{E}_{y \sim \pi(\cdot|x)}[r(x, y)] - \tau^L \mathrm{KL}_x(\pi \| \pi^{\mathrm{ref}}) \right]$$

which is again a KL-regularized optimization problem that admits a closed-form solution

$$\pi^{\star}(y \mid x) = \frac{1}{Z(x)} \pi^{\mathrm{ref}}(y \mid x) \exp\left(\tfrac{1}{\tau^L} r(x, y)\right).$$

$\square$

### A.2 Proof of Theorem 2

*Proof.* This lemma is folklore in the algorithmic game theory community and can be quickly verified.

Let $x \in \mathcal{X}$. Given any action $y \in \mathcal{Y}$, there exists a not necessarily unique $y' \in \mathcal{Y}$ minimizing $p(y \succ y' \mid x)$. Hence, irrespective of the Leader's policy $\pi(\cdot \mid x)$, there always exists a Follower's deterministic best response policy $\omega_{\mathrm{br}}(\cdot \mid x, y)$ with $\omega_{\mathrm{br}}(y' \mid x, y) = 1$ for some $y'$. In other words, the Follower always has a deterministic best response policy.

Similarly, the optimization problem for the Leader given some context $x$ reduces to finding $y$ that maximizes $\mathbb{E}_{y' \sim \omega_{\mathrm{br}}(\cdot \mid x, y)}[p(y \succ y' \mid x)]$ so that the SLHF optimization problem admits a determinsitic solution. $\square$

### A.3 Concave-Convex Property of $f$

We show here that the objective function $f$ in Equation (5) of the Stackelberg optimization problem is concave-convex. Similar results were established in the context of NLHF by Munos et al. (2024).

Throughout this section, we assume $|X| = 1$ and omit $x$ from the notation for clarity. All results extend directly to the general case with a finite context space $\mathcal{X}$.

Then, the objective function of Equation (5) is given by

$$f(\pi, \omega) = \mathbb{E}_{y \sim \pi(\cdot), y' \sim \omega(\cdot \mid y)}\big[p(y \succ y')\big] - \tau^L \mathrm{KL}\big(\pi \,\|\, \pi^{\mathrm{ref}}\big) + \tau^F \mathbb{E}_{y \sim \pi(\cdot)}[\mathrm{KL}_y\big(\omega \,\|\, \omega^{\mathrm{ref}}\big)]. \quad (9)$$

The first term is bilinear in $\pi$ and $\omega$, as shown by expanding the expectation:

$$\mathbb{E}_{y \sim \pi(\cdot), y' \sim \omega(\cdot \mid y)}\big[p(y \succ y')\big] = \sum_{y \in \mathcal{Y}} \pi(y) \sum_{y' \in \mathcal{Y}} p(y \succ y') \omega(y' \mid y).$$

The KL terms are convex in their respective arguments. Hence, $f$ is bilinear when $\tau^L = \tau^F = 0$.

### A.4 Comments on the Convergence of StackelbergGDA (Algorithm 1)

We here want to briefly comment on the ergodic convergence guarantee of StackelbergGDA that is a consequence of well-known results in the literature. We also briefly discuss limitations and future work to achieve better theoretical convergence guarantees by adapting existing ideas from the optimization and NLHF literature to the SLHF problem.

As previously shown, the SLHF objective $f(\pi, \omega)$ is concave-convex. It is well-known that two-timescale GDA then converges in the ergodic sense, i.e., the averaged iterates converge to the equilibrium with gradient complexity $\mathcal{O}(\varepsilon^{-2})$ (Nedić and Ozdaglar, 2009). In the strongly-concave-strongly-convex setting, standard results also tell us that two-timescale GDA converges in the last-iterate with complexity $\mathcal{O}(\kappa^2 \log \frac{1}{\varepsilon})$ (Zhang et al., 2022; Zamani et al., 2024; Lin et al., 2024). Unfortunately, the KL divergence is not strongly convex w.r.t. the $\ell_2$ norm so that these results do not directly apply, and deriving linear last-iterate guarantees for StackelbergGDA is challenging. In future work, it will be interesting to analyze whether, e.g., extragradient (Zhou et al., 2025) or mirror descent (Munos et al., 2024) approaches that have been successfully applied to NLHF, can be adapted to the SLHF framework to guarantee fast last-iterate convergence.

## B Scalable Implementation of StackelbergGDA

When fine-tuning large language models, the context and action spaces $\mathcal{X}$ and $\mathcal{Y}$ are far too large to optimize over $\Pi$ and $\Omega$ directly. To address this, we introduce a practical variant of StackelbergGDA in Algorithm 2.

**Policy Parameterization.** We replace the tabular policies $\pi$ and $\omega$ with neural parameterizations $\pi_\theta$ and $\omega_\phi$ (e.g., transformer networks). This renders the policy spaces tractable via their parameter vectors $\theta$ and $\phi$, however, the concave-convex property does not necessarily carry over to the parameters $\theta$ and $\phi$.

**Batched, Variance-reduced Gradient Estimates.** Exact evaluation of the expectations in $\nabla f$ is infeasible due to the expectation over the context and action spaces. Instead, at each iteration we sample a batch of size $B$:

$$\{(x_i, y_i, y_i', p_i)\}_{i=1}^B, \ x_i \sim \rho, \ y_i \sim \pi_\theta(\cdot \mid x_i), \ y_i' \sim \omega_\phi(\cdot \mid x_i, y_i), \ p_i = p(y_i \succ y_i' \mid x_i).$$

We then form unbiased estimates as

$$\widehat{\nabla}_\theta f = \frac{1}{B} \sum_{i=1}^B (p_i - \tau^L k_i^L) \nabla_\theta \log \pi_\theta(y_i \mid x_i), \widehat{\nabla}_\phi f = \frac{1}{B} \sum_{i=1}^B (p_i - \tau^F k_i^F) \nabla_\phi \log \omega_\phi(y_i' \mid x_i, y_i),$$

with likelihood ratios $k_i^L = \frac{\pi_\theta(y_i|x_i)}{\pi^{\mathrm{ref}}(y_i|x_i)}$ and $k_i^F = \frac{\omega_\phi(y_i'|x_i,y_i)}{\omega^{\mathrm{ref}}(y_i'|x_i,y_i)}$. The derivation of these gradients follow the policy gradient method described in Williams (1992). The gradient estimators are naturally compatible with additional variance reduction techniques such as subtracting a constant baseline.

**Single-Model Instantiation.** Simultaneously training two billion-parameter transformer models is memory-prohibitive. Similarly to SCORE (Kumar et al., 2025), we collapse both Leader and Follower into one model $\pi_\theta$ by using distinct chat templates (Figure 1). When the model is only given the context $x$, we use the template in Figure 1(a) that only includes $x$ as the prompt. When the model is given both the context $x$ and an action $y$, we use the template in Figure 1(b) that includes both the context $x$ and the action $y$, as well as a predefined instruction to improve the action $y$.

Then, letting $\kappa = \frac{\alpha^F}{\alpha^L}$ denote the two-timescale weight coefficient, we optimize the model to minimize the following loss function

$$\mathcal{L}(\theta) = -\frac{1}{B} \sum_{i=1}^B (p_i - \tau^L k_i^L) \log \pi_\theta(y_i \mid x_i) + \frac{\kappa}{B} \sum_{i=1}^B (p_i - \tau^F k_i^F) \log \pi_\theta(y_i' \mid x_i, y_i). \quad (10)$$

Gradient steps on $\mathcal{L}(\theta)$ realize the two-time-scale gradient descent-ascent updates via a single network, thereby substantially reducing memory usage.

---

**Algorithm 2** STACKELBERGGDA (Practical)

1: **procedure** STACKELBERGGDA$(\mathcal{X}, \mathcal{Y}, \rho, \eta)$
2:     Initialize the parameter $\theta$ for the shared model
3:     **for** $i = 1, 2, \ldots$ **do**
4:         **for** $b = 1, \ldots, B$ **do**
5:             Sample prompt $x_b \sim \rho$
6:             Sample Leader response using the prompt in Figure 1(a): $y_b \sim \pi_\theta(\cdot \mid x_b)$
7:             Sample Follower response using the prompt in Figure 1(b): $y_b' \sim \pi_\theta(\cdot \mid x_b, y_b)$
8:             Observe preference feedback $p_b = p(y_b \succ y_b' \mid x_b)$
9:         **end for**
10:         Update the weights $\theta$ according to the loss in Equation (10): $\theta \leftarrow \theta - \eta \nabla_\theta \mathcal{L}(\theta)$
11:     **end for**
12: **end procedure**

---

**Computational Comparison.** At training time, the computational cost of STACKELBERGGDA is comparable to standard online RLHF/NLHF algorithms, with only marginal overhead from the Follower's prompt. Specifically, STACKELBERGGDA requires two samples per prompt, matching most NLHF methods and remaining more efficient than algorithms like NASH-MD-PG (Munos et al., 2024) or ONLINEIPO (Calandriello et al., 2024) that rely on expensive mixture policies. While SPO (Swamy et al., 2024) uses only one sample, its efficacy on LLM-scale tasks is not yet explored. Finally, compared to RLHF, STACKELBERGGDA avoids the high memory cost of PPO (Schulman et al., 2017) which is due to the value function estimation. Recent memory-efficient RLHF methods, such as RLOO (Ahmadian et al., 2024) and GRPO (Shao et al., 2024), only resolve this memory issue by introducing sampling costs comparable to, or even higher than, STACKELBERGGDA.

## C  IMPLEMENTATION DETAILS

**Implementation Codebase.**  We trained RLOO[5] and NASH-MD-PG[6] using their implementations in the TRL Python package (von Werra et al., 2020). For all training runs, including reward modeling, we used Low-Rank Adaptation (LoRA) (Hu et al., 2022) with rank $r = 32$, scaling factor $\alpha = 64$, and dropout rate set to 0.1. All code used for the results presented in Section 6 is available at: https://github.com/lasgroup/stackelberg-learning.

**Compute Resources.**  Experiments in Section 6.1 were conducted on a single node with 8 Nvidia GeForce RTX 4090 GPUs. The total compute time, including hyperparameter sweeps, was approximately 4,000 GPU-hours. The training run in Section 6.2 was conducted on a single node with 4 Nvidia GH200 GPUs using approximately 1,300 GPU-hours.

## D  ADDITIONAL EXPERIMENTAL RESULTS

### D.1  PREFERENCE MODEL

We estimate the preference model used to fine-tune the LLMs by treating the five attributes in the HELPSTEER2 datasets (Wang et al., 2024) as distinct annotators, denoted by the set $\mathcal{A} = \{helpfulness, correctness, coherence, complexity, verbosity\}$, and define $\nu$ as a uniform distribution over $\mathcal{A}$. For each attribute $a \in \mathcal{A}$, we estimate a reward function $\hat{r}_a$ using the Bradley-Terry model and maximize the log-likelihood on the training dataset $\mathcal{D} = \{(x_i, y_i^w, y_i^l)\}_{i=1}^N$

$$\min_r \sum_{i=1}^N \sigma(r(x_i, y_i^w) - r(x_i, y_i^l)) + \lambda(r(x_i, y_i^w) + r(x_i, y_i^l))^2.$$

We here decided which response is the winning and losing one in the dataset by comparing the attribute scores provided by the annotators. The additional regularization ensures that the rewards are centralized around zero (Eisenstein et al., 2024). For the attributes correctness, helpfulness, and coherence, we consider higher scores to be better while for verbosity and complexity lower values are more preferable. This is in accordance with the scoring criteria described in Wang et al. (2024). Each reward function is trained independently, initialized from the QWEN2.5-1.5B[7] model with a single linear head. We train each model for 5 epochs on the training prompts and completions with batch size 32, learning rate $1e-4$, and regularization coefficient $\lambda = 0.01$. The final accuracies of the models on the validation dataset are $78\%, 65\%, 61\%, 60\%$, and $59\%$ for verbosity, complexity, correctness, helpfulness, and coherence, respectively.

The overall preference function $p$ is then defined as

$$p(y \succ y' \mid x) = \frac{1}{|\nu|} \sum_{a \in \nu} \mathbf{1}\{\hat{r}_a(x, y) \geq \hat{r}_a(x, y')\}. \tag{11}$$

We evaluate the non-transitivity of the preference model $p$ defined in Equation (11) on the validation prompts and five responses from each of the four models used for comparison in Section 6. For each prompt, we construct a complete directed graph between the 20 completions as nodes and edges directed from the non-preferred completion towards the preferred one. Figure 2 illustrates this directed graph on the first prompt of the validation dataset. $57\%$ of the directed graphs include cycles, which illustrate the intransitivity of the preference function $p$ on the completion space $\mathcal{Y}$. Furthermore, there are almost 19 million cycles in the dataset across all prompts. From these cycles, $5.79\%$ is length 9 or shorter, $41.64\%$ is between length 10 and 12, $42.72\%$ is of length 13 or 14, and $9.85\%$ is longer. Also, $33.31\%$ of the prompts have at least one but no more than 9 cycles, $17.12\%$ has between 10 and 99, $3.63\%$ has between 100 and 999, and $2.54\%$ has at least 1000.

---

[5]https://huggingface.co/docs/trl/main/en/rloo_trainer
[6]https://huggingface.co/docs/trl/main/en/nash_md_trainer
[7]https://huggingface.co/unsloth/Qwen2.5-1.5B-Instruct

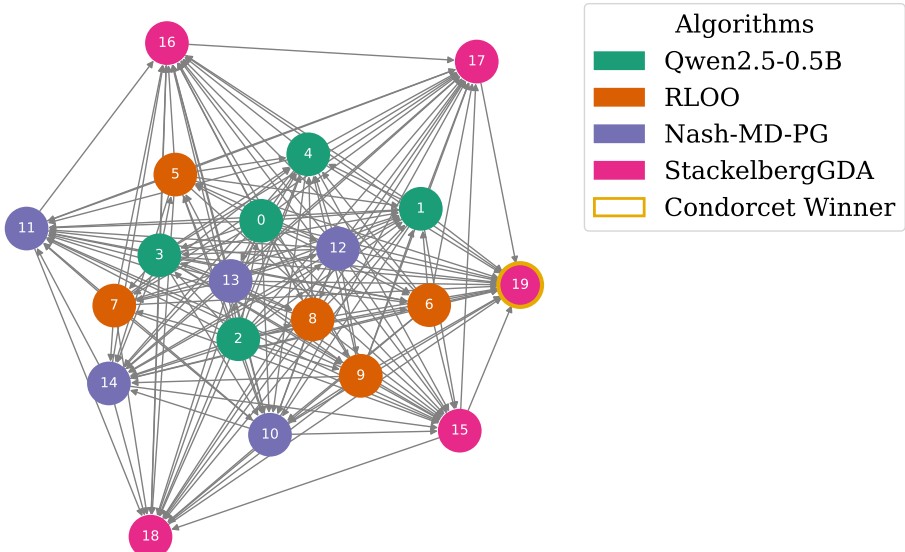

Figure 2: Directed graph based with completions generated by the fine-tuned models and edge directions representing the preference between them.

## D.2 ITERATIVE IMPROVEMENTS AT TEST-TIME

We extend the experimental results from Section 6 by analyzing iterative improvements and performance scaling with increased test-time computation. Our results suggests that with increasing test-time computation the benefit of fine-tuning using both RLOO or NASH-MD-PG is negligible compared to using the base model QWEN2.5-0.5B. In contrast, STACKELBERGGDA yields strict improvements. Building on the example in Section 4.1, we assume that at test-time, a single annotator $a \sim \nu$ and a context $x \sim \rho$ are sampled.

For the base model QWEN2.5-0.5B and the models with fine-tuned with RLOO and NASH-MD-PG, we independently sample $N$ responses $y_1, \ldots, y_N$. For STACKELBERGGDA, which inherently supports iterative refinement, we generate the first sample from the Leader policy $y_1 \sim \pi^\star(\cdot \mid x)$, and subsequent responses from the Follower policy $y_i \sim \omega^\star(\cdot \mid x, y_{i-1})$ for $i \geq 2$. We define $y_{1:N} \coloneqq (y_1, \ldots, y_N)$.

In line with prior work on Best-of-$N$ sampling (Openai et al., 2023; Beirami et al., 2024; Dubois et al., 2024; Sessa et al., 2024), we evaluate the quality of the $N$ samples by computing the maximum reward obtained for each attribute under the sampled annotator's reward model, that is,

$$\hat{r}_a^N(x, y_{1:N}) \coloneqq \max_{y_1, \ldots, y_N} \hat{r}_a(x, y_i). \tag{12}$$

Analogous to the preference function $p$ defined in Section 3, in this section, we compare two models $\pi$ and $\pi'$ w.r.t. the preference functions derived from the annotator-specific reward functions under Best-of-$N$ sampling:

$$p_a^N(\pi \succ \pi' \mid x) \coloneqq \mathbb{E}_{\substack{y_{1:N} \sim \pi(\cdot \mid x) \\ y'_{1:N} \sim \pi'(\cdot \mid x)}} \left[ \mathbb{1}\{\hat{r}_N^a(x, y_{1:N}) \geq \hat{r}_N^a(x, y'_{1:N})\} \right]. \tag{13}$$

**Notational Note.** We here adopt a slight abuse of notation. Specifically, we write $y_{1:N} \sim \pi(\cdot \mid x)$ to denote the sampling of $N$ responses from a model $\pi$, even though this notation does not faithfully represent the sampling procedure used by STACKELBERGGDA. For QWEN2.5-0.5B, RLOO, and NASH-MD-PG, the samples $y_1, \ldots, y_N$ are drawn i.i.d. from a single model $\pi(\cdot \mid x)$. In contrast, for STACKELBERGGDA, the sampling process is inherently autoregressive: we first draw $y_1 \sim \pi^\star(\cdot \mid x)$ from the Leader policy, and then generate $y_i \sim \omega^\star(\cdot \mid x, y_{i-1})$ for $i \geq 2$ using the Follower policy. Despite this difference, we overload the notation $y_{1:N} \sim \pi(\cdot \mid x)$ to unify the presentation in

Table 6: Preference scores for RLOO versus QWEN2.5-0.5B across all attributes as a function of the number of test-time samples $N$.

| Attribute | Number of Samples $N$ | | | | |
|---|---|---|---|---|---|
| | 1 | 2 | 3 | 4 | 5 |
| Coherence | 0.521 | 0.338 | 0.262 | 0.207 | 0.164 |
| Complexity | 0.892 | 0.842 | 0.801 | 0.771 | 0.754 |
| Correctness | 0.388 | 0.212 | 0.139 | 0.098 | 0.070 |
| Helpfulness | 0.322 | 0.150 | 0.087 | 0.057 | 0.036 |
| Verbosity | 0.846 | 0.777 | 0.728 | 0.695 | 0.669 |
| **Average** | **0.594** | **0.464** | **0.403** | **0.366** | **0.338** |

Equations (12) and (13). In the case of STACKELBERGGDA, this notation should be interpreted as shorthand for the autoregressive sampling process described above.

Previous work has shown that Best-of-$N$ sampling can rival the performance of RLHF-based fine-tuning (Dubois et al., 2023; Sessa et al., 2024; Beirami et al., 2024). Motivated by this, we compare the preference scores defined in Equation (13) of RLOO, NASH-MD-PG, and STACKELBERGGDA with respect to the base model QWEN2.5-0.5B. Throughout this section, we consider the maximum number of samples to be $N = 5$.

**Results.** Table 6 reports the preference scores for RLOO. While the model initially (i.e. $N = 1$) performs competitively on complexity and verbosity attributes, iterative sampling reveals a collapse into a single preference mode. In particular, we observed deterministic outputs for RLOO the generic response: *"I apologize, but I'm unable to engage in conversations about political topics. If you have any other questions or need further assistance with a different subject, feel free to ask."* As a result, the model's diversity and coverage deteriorate, and its overall preference scores (relative to QWEN2.5-0.5B) decline sharply as $N$ increases.

In contrast, NASH-MD-PG demonstrates some benefit from additional sampling, as shown in Table 7. Its preference score for verbosity remains stable and it shows moderate improvement in coherence. However, for the remaining attributes (correctness, helpfulness, and complexity) its gains are slower than those achieved by QWEN2.5-0.5B with Best-of-$N$ sampling. Consequently, the overall preference score of NASH-MD-PG declines with increasing $N$, suggesting that the model fine-tuned with NASH-MD-PG does not improve notably (compared to the base model) when the number of samples drawn at test-time increases.

On the other hand, STACKELBERGGDA exhibits more favorable behavior. As shown in Table 8, while the preference score on verbosity and complexity taper off with more samples, STACKELBERGGDA achieves notably faster gains on coherence, correctness, and helpfulness. For these attributes, the preference rate improves by 10 percentage points or more, making STACKELBERGGDA the only method among the three to demonstrate consistent improvement over QWEN2.5-0.5B as $N$ increases. This means that the performance of STACKELBERGGDA effectively scales with test-time compute.

The strong emphasis on complexity and verbosity by RLOO is expected, as it optimizes the average reward across all five attributes, and these two dimensions yield the highest values. However, for NASH-MD-PG, this outcome is less expected. We hypothesize that it stems from its training objective, which pits the policy against a mixture of the reference policy QWEN2.5-0.5B and the most recent iteration. Once NASH-MD-PG outperforms QWEN2.5-0.5B on all attributes, it begins focusing on attributes where further improvement over itself is possible, namely, complexity and verbosity. Nevertheless, this skewed emphasis is suboptimal: annotators prefer models that perform well on all attributes. In fact, a policy that focuses on coherence, correctness, and helpfulness is preferred by $60\%$ of the annotators. STACKELBERGGDA's asymmetric formulation that trains a Leader and a Follower policy separately (though potentially unified in a single model) helps mitigate this imbalance across attributes. This leads to a more balanced policy that is preferred by a wider range of annotators.

Table 7: Preference scores for NASH-MD-PG versus QWEN2.5-0.5B across all attributes as a function of the number of test-time samples $N$.

| Attribute | Number of Samples $N$ | | | | |
|---|---|---|---|---|---|
| | 1 | 2 | 3 | 4 | 5 |
| Coherence | 0.731 | 0.743 | 0.757 | 0.763 | 0.760 |
| Complexity | 0.761 | 0.743 | 0.732 | 0.726 | 0.727 |
| Correctness | 0.633 | 0.615 | 0.620 | 0.617 | 0.615 |
| Helpfulness | 0.645 | 0.633 | 0.640 | 0.641 | 0.638 |
| Verbosity | 0.858 | 0.855 | 0.850 | 0.849 | 0.852 |
| **Average** | **0.726** | **0.718** | **0.720** | **0.719** | **0.718** |

Table 8: Preference scores for STACKELBERGGDA versus QWEN2.5-0.5B across all attributes as a function of the number of test-time samples $N$.

| Attribute | Number of Samples $N$ | | | | |
|---|---|---|---|---|---|
| | 1 | 2 | 3 | 4 | 5 |
| Coherence | 0.778 | 0.850 | 0.865 | 0.875 | 0.873 |
| Complexity | 0.714 | 0.666 | 0.628 | 0.600 | 0.592 |
| Correctness | 0.670 | 0.762 | 0.791 | 0.795 | 0.803 |
| Helpfulness | 0.692 | 0.767 | 0.783 | 0.786 | 0.791 |
| Verbosity | 0.833 | 0.820 | 0.798 | 0.777 | 0.765 |
| **Average** | **0.738** | **0.773** | **0.773** | **0.767** | **0.765** |

### D.3 ABLATION ON THE TWO-TIMESCALE COEFFICIENT

Table 9 and Table 10 present ablations on the follower weight parameter $\kappa$ in STACKELBERGGDA's loss function (10) when fine-tuning the QWEN2.5-0.5B and QWEN2.5-1.5B models, respectively. Each row reports the average preference scores over the corresponding initial policy, for both the Leader and Follower policies, on the training and validation splits. These results highlight the importance of balancing the two components of STACKELBERGGDA's asymmetric training objective. In general, moderate values of $\kappa$ can help the Follower improve without compromising the Leader too severely, but excessively large weights may impair both players.

In Table 9, we observe that increasing $\kappa$ leads to a gradual decline in the Leader's performance. While the Follower benefits from increasing $\kappa$ from 1 to 5, performance worsens at $\kappa = 10$ for both the Leader and Follower, indicating an overemphasis on the Follower's loss can destabilize the overall optimization.

Table 10 shows a similar trend for the larger QWEN2.5-1.5B model. Due to the decrease of performance above $\kappa = 5$ in Table 9, we carry out the ablation on a finer grid $\kappa \in \{1, 2, 3, 4, 5\}$. Moreover, we evaluate each model after 2000 training steps as a larger base model requires more gradient updates to converge. While the performance of $\kappa = 1$ stands out in Table 10, we observe that it is overfitting to verbosity and complexity by responding to every prompt with short, non-informative answers asking for further information such as *"Certainly! If you need detailed insights on technical topics like that, feel free to ask—I'm here to assist with informatively aligned queries!"*. On the contrary to the collapse observed for RLOO in Section D.2, the model remains stochastic with the responses having similar information content. This outcome demonstrates the effectiveness of STACKELBERGGDA in optimizing its objective despite the qualitatively undesirable responses.

### D.4 MODEL SCALING

We extend our round-robin comparison from Section 6.1.1 to larger models within the Qwen2.5 family, specifically, QWEN2.5-1.5B and QWEN2.5-3B (Qwen et al., 2024). These evaluations demonstrate that STACKELBERGGDA continues to be on par or outperform baselines even as model

Table 9: Ablation on the follower weight parameter $\kappa$ in STACKELBERGGDA's loss function (10) fine-tuning the QWEN2.5-0.5B model. Scores show the average preference over the base model.

| Follower Weight $\kappa$ | Train | | Validation | |
|---|---|---|---|---|
| | Leader | Follower | Leader | Follower |
| 1 | **0.768** | 0.804 | **0.761** | 0.784 |
| 5 | 0.743 | **0.814** | 0.723 | **0.806** |
| 10 | 0.725 | 0.800 | 0.710 | 0.783 |

Table 10: Ablation on the follower weight parameter $\kappa$ in STACKELBERGGDA's loss function (10) fine-tuning the QWEN2.5-1.5B model. Scores show the average preference over the base model.

| Follower Weight $\kappa$ | Train | | Validation | |
|---|---|---|---|---|
| | Leader | Follower | Leader | Follower |
| 1 | **0.848** | **0.852** | **0.850** | **0.851** |
| 2 | 0.718 | 0.737 | 0.719 | 0.730 |
| 3 | 0.767 | 0.806 | 0.771 | 0.803 |
| 4 | 0.733 | 0.811 | 0.736 | 0.807 |
| 5 | 0.720 | 0.819 | 0.720 | **0.818** |

size increases. Since larger models require more training updates to reach convergence in our setup, we train NASH-MD-PG for 1,500 steps and STACKELBERGGDA for 2,000 steps. The RLOO method converges earlier and requires only 1,000 steps even for these larger models. We fix the follower-weight parameter at $\kappa = 5$ for both scales, based on our ablation results in Section D.3.

Table 11 summarizes results for models fine-tuned from QWEN2.5-1.5B. Both NASH-MD-PG and STACKELBERGGDA clearly outperform the base model and the RLOO baseline. While the Leader policy of STACKELBERGGDA underperforms compared to NASH-MD-PG, the Follower policy conditioned on the Leader's responses matches or exceeds NASH-MD-PG's performance, mirroring the improvements observed when starting from the QWEN2.5-0.5B in Table 3. As noted in Section D.3, this performance gap between the Leader and NASH-MD-PG could likely be reduced by tuning $\kappa$, albeit at the potential cost of Follower quality.

Table 12 shows analogous comparisons for models initialized from QWEN2.5-3B. Here, STACKEL-BERGGDA again performs strongly, with its Follower policy matching or surpassing NASH-MD-PG across most pairwise matchups, and both algorithms outperforming the base model. NASH-MD-PG and STACKELBERGGDA are closely matched when compared directly. Due to compute limitations, we capped training at 2,000 steps for these larger models. Nonetheless, the Leader policy continued to improve near the end of training, suggesting further gains in preference score may be possible with additional updates.

Table 11: Pairwise preference comparisons between the responses of QWEN2.5-0.5B, QWEN2.5-1.5B, RLOO, NASH-MD-PG, and STACKELBERGGDA algorithms. Fine-tuned models are trained from the QWEN2.5-1.5B. Each cell shows the average preference score of the row model over the column model.

| | QWEN2.5-0.5B | QWEN2.5-1.5B | RLOO | NASH-MD-PG | STACKELBERGGDA | |
| --- | --- | --- | --- | --- | --- | --- |
| | | | | | LEADER | FOLLOWER |
| QWEN2.5-0.5B | 0.000 | 0.479 | 0.379 | 0.188 | 0.271 | 0.166 |
| QWEN2.5-1.5B | 0.521 | 0.000 | 0.401 | 0.209 | 0.293 | 0.187 |
| RLOO | 0.621 | 0.599 | 0.000 | 0.197 | 0.310 | 0.175 |
| NASH-MD-PG | 0.812 | 0.791 | 0.803 | 0.000 | 0.623 | 0.489 |
| STACKELBERGGDA LEADER | 0.729 | 0.707 | 0.690 | 0.377 | 0.000 | 0.313 |
| STACKELBERGGDA FOLLOWER | **0.834** | **0.813** | **0.825** | **0.511** | **0.687** | 0.000 |

Table 12: Pairwise preference comparisons between the responses of QWEN2.5-0.5B, QWEN2.5-3B, RLOO, NASH-MD-PG, and STACKELBERGGDA algorithms. Fine-tuned models are trained from the QWEN2.5-3B. Each cell shows the average preference score of the row model over the column model.

| | QWEN2.5-0.5B | QWEN2.5-3B | RLOO | NASH-MD-PG | STACKELBERGGDA | |
| --- | --- | --- | --- | --- | --- | --- |
| | | | | | LEADER | FOLLOWER |
| QWEN2.5-0.5B | 0.000 | 0.504 | 0.399 | 0.187 | 0.304 | 0.187 |
| QWEN2.5-3B | 0.496 | 0.000 | 0.412 | 0.199 | 0.319 | 0.179 |
| RLOO | 0.601 | 0.588 | 0.000 | 0.173 | 0.338 | 0.201 |
| NASH-MD-PG | **0.813** | **0.801** | **0.827** | 0.000 | **0.638** | **0.507** |
| STACKELBERGGDA LEADER | 0.696 | 0.681 | 0.662 | 0.362 | 0.000 | 0.312 |
| STACKELBERGGDA FOLLOWER | **0.813** | **0.821** | **0.799** | **0.493** | **0.688** | 0.000 |

