# OpenReview forum: "Stackelberg Learning from Human Feedback: Preference Optimization as a Sequential Game"
_ICLR.cc/2026/Conference — ICLR 2026 Poster_

### Official Review · Reviewer_Uhos · 2025-10-30

**Soundness:** 2
**Presentation:** 2
**Contribution:** 3
**Rating:** 4
**Confidence:** 3

**Summary:**

This paper introduces Stackelberg Learning from Human Feedback (SLHF), a framework for preference optimization that models alignment as a sequential-move game between a Leader policy and a Follower policy. The authors propose StackelbergGDA, a two-timescale gradient descent-ascent algorithm to approximate the Stackelberg equilibrium. The framework naturally enables inference-time refinement through iterative sampling. Experiments on language models ranging from 0.5B to 8B parameters demonstrate that SLHF achieves strong alignment performance, with the Follower policy consistently outperforming RLHF and NLHF baselines.

**Strengths:**

- **Strong motivation and intuition:** The paper provides excellent motivating examples, particularly the Condorcet paradox analysis in Section 4.1, which clearly illustrates how SLHF, RLHF, and NLHF differ in handling intransitive preferences.

- **Practical value of inference-time refinement:** The Leader-Follower structure naturally supports inference-time improvement without additional training, which is valuable for LLM applications. The empirical results in Table 4 demonstrate impressive cross-model generalization.

- **Comprehensive experiments:** The paper includes thorough empirical evaluation across multiple model scales (0.5B to 8B parameters) and provides detailed ablations and additional results in the appendix.

**Weaknesses:**

### Major Weaknesses
1. **Mischaracterization and Missing Literature on NLHF**

**Issue:** The claim in the introduction that "simultaneous play forces both players to optimize against a moving opponent which can hinder convergence" is not accurate. Recent NLHF works have established polynomial or even linear convergence rates to Nash equilibrium: https://arxiv.org/abs/2312.00886, https://arxiv.org/abs/2401.04056, https://arxiv.org/abs/2410.16714, https://arxiv.org/abs/2503.08942. The paper should acknowledge these theoretical convergence guarantees rather than suggesting NLHF inherently struggles with convergence.

**Critical omission:** A closely related work (https://arxiv.org/abs/2502.18099v2) that also studies Stackelberg games for LLM alignment was published over half a year ago and cannot be considered concurrent work. This significantly undermines the novelty claim. The authors must thoroughly discuss this work and clarify their contributions relative to it.

2. **Problematic Characterization of Mixed Strategies**

**Issue 1:** At the end of Section 3, the authors state: "when no action is majority-preferred the equilibrium necessarily involves mixed strategies. This inherent stochasticity can be undesirable in applications where consistency and reliability are critical."
This characterization is misleading for several reasons: (1) In RLHF/NLHF/SLHF with KL regularization ($\tau > 0$), the optimal policy is always a distribution over responses, not deterministic. (2) A "deterministic policy" in LLM is not well-defined, unless we sample with temperature 0, but the model still outputs a probability distribution.

**Issue 2:** The claim before Section 4.1 that "there exists a deterministic Stackelberg equilibrium" suffers from the same conceptual problem. With regularization (which the paper uses throughout), policies must be stochastic. The best response is essentially the RLHF solution when viewing win-rate as reward.

3. **Lack of Theoretical Analysis:** The paper provides no convergence guarantees for either Algorithm 1 or Algorithm 2. Key questions remain unanswered: Does StackelbergGDA converge to a Stackelberg equilibrium? What is the convergence rate? Under what conditions does convergence occur? Given that RLHF/NLHF methods now have established convergence theory, the lack of any theoretical analysis for SLHF is a significant weakness. The paper should at minimum discuss the challenges in proving convergence or provide experimental evidence of convergence behavior.

4. **Algorithm Presentation Issues:** Inconsistency: Algorithms 1 and 2 are essentially different algorithms (analogous to OGDA vs. OMWU, https://arxiv.org/abs/2006.09517). For consistency and clarity, I suggest replacing Algorithm 1 with a theoretical version of Algorithm 2.

### Minor Weaknesses and Questions

5. **Baseline choice:** The paper uses Nash-MD-PG as the primary NLHF baseline, which has been shown to converge extremely slowly (see Section 5 of https://arxiv.org/abs/2503.08942). How would SLHF compare against faster NLHF algorithms like those mentioned above?

6. The intransitivity analysis (57% of graphs contain cycles) is interesting but could be expanded—what is the typical cycle length? How does this compare to other datasets?

**Questions:**

Please address my concerns in the Weakness section. There is no other questions.

---

> ### Author Response · Authors · 2025-11-17
> **Answer Part 1**
>
> Dear Reviewer Uhos,
>
> Thank you for your time and effort in reviewing our work and providing helpful feedback. We appreciate you highlighting the strong motivation and intuition of our work, the practical value of inference-time refinement, and our comprehensive experiments.
>
> We would like to address your concerns and questions below.
> 1. **Mischaracterization and Missing Literature on NLHF**
> We agree that our statement in the introduction was inaccurate, and we have removed it. We have also incorporated the missing references (e.g., EGPO, SGPO) into our related work section (cf. updated submission), and highlighting the convergence guarantees of, e.g., EGPO. Regarding SGPO (Xu et al., 2025), we appreciate the pointer and now discuss it explicitly in the main text (Lines 90-93). Though similar in name, SGPO studies a Stackelberg game between a policy and an adversarial preference model, which is conceptually more closely related to (Shen et al., 2024; Thoma et al., 2024; Makar-Limanov et al., 2024). In contrast, our work formulates a Stackelberg game between two policies directly over a pairwise preference model, without assuming transitivity or an adversarial reward. We have clarified this distinction in the revised related work section.
>
> 2. **Problematic Characterization of Mixed Strategies**
> We agree that with KL regularization ($\tau > 0$) the optimal policies in RLHF, NLHF, and SLHF are necessarily stochastic. Our intention was to highlight that, in the unregularized setting ($\tau = 0$), the Nash solution must mix under cyclic preferences, whereas the Stackelberg solution admits a deterministic equilibrium (Remark 2). We agree that this distinction would be clearer if we explicitly noted in Section 3 that this is meaningful only for $\tau = 0$, and we have revised the text accordingly (revisions highlighted in color). You are also right that deterministic policies are defined for LLMs only when greedy sampling is used. Please note, however, that we deliberately introduce SLHF in Section 4 using a general game-theoretic formulation without yet focusing only on LLMs.
>
> 3. **Lack of Theoretical Analysis**
> We agree that providing convergence guarantees for StackelbergGDA is an important direction. As discussed in Section 5 (second last paragraph), the SLHF objective is concave-convex (Appendix A), and classical results imply that the averaged iterates of two-timescale GDA converge to a Stackelberg equilibrium at the standard $\mathcal{O}(\varepsilon^{-2})$ rate. However, obtaining strong last-iterate convergence, as recent NLHF work such as EGPO has done, is more challenging, and two-timescale GDA is known to guarantee last-iterate convergence only in the strongly-convex-strongly-concave setting, which does not hold for our objective (the KL is not strongly convex in the $\ell_2$-norm). Developing SLHF algorithms with provable last-iterate convergence is an important open problem, and we have added a dedicated discussion of these challenges in Appendix A.4 and emphasized this point in the limitations in Section 7. Our focus in this paper is proposing the SLHF framework, analyzing its properties, introducing inference-time refinement, and providing a thorough experimental evaluation of SLHF compared to RLHF and NLHF.
>
> 4. **Algorithm Presentation Issues**
> We would like to ask you to elaborate on his comment about Algorithms 1 and 2, and describe the inconsistencies you refer to. We agree that there are modifications between the algorithms due to the implementation details we describe in Appendix B. However, the gradients $\nabla_{\theta} f$ and $\nabla_{\phi} f$ are derived from the gradient of the objective function $f$’s stochastic approximation w.r.t. the model parametrization weights following the GDA algorithm. We clarified in Appendix B that this gradient is based on policy gradient derivation of the REINFORCE algorithm introduced by Williams (1992). We also incorporate the difference in the learning rates in the loss function $\mathcal{L}$ for the shared parametrization using the scaling factor $\kappa$. Therefore, we could not identify major inconsistencies.

---

> > ### Author Response · Authors · 2025-11-17
> > **Answer Part 2**
> >
> > 5. **Baseline choice**
> > Our choice for comparison algorithms in Section 6.1 was based on three factors, which we have now clarified in Section 6:
> >
> > * Framework comparison: Our primary goal was to compare the solution concepts (RLHF vs. NLHF vs. SLHF), not algorithm runtime comparison.
> >
> > * Conceptual similarities: We chose RLOO and Nash-MD-PG as representative algorithms for RLHF and NLHF as they are also building on policy gradient updates similarly to StackelbergGDA.
> >
> > * Implementation Parity: The TRL library includes well-tested and reliable implementations of the chosen algorithms. By building StackelbergGDA similarly within TRL, we minimize performance differences due to implementation-level "tricks" and ensure a fairer comparison of the underlying algorithms.
> >
> >   For completeness, we ran experiments with EGPO (Zhou et al. 2025) on the 0.5B and 1.5B model scales as they provide an open-source implementation in TRL. We set the learning rate and KL regularization coefficient the same as for the other algorithms, and keep the rest to their default values. Our results in the following tables (Extension of Table 3 and 11 in the main text) show that EGPO underperforms compared to Nash-MD-PG and Stackelberg GDA and only exceeds RLOO on the 0.5B scale, therefore, we consider Nash-MD-PG a suitable representative algorithm for NLHF.
> >
> > | | Qwen2.5-0.5B | RLOO | Nash-MD | Nash-EGPO | Stackelberg-PG-Leader | Stackelberg-PG-Follower |
> > | :--- | :--- | :--- | :--- | :--- | :--- | :--- |
> > | **Qwen2.5-0.5B** | 0.0000 | 0.407 | 0.279 | 0.466 | 0.2664 | 0.200 |
> > | **RLOO** | 0.592 | 0.000 | 0.393 | 0.528 | 0.387 | 0.344 |
> > | **Nash-MD** | 0.721 | 0.607 | 0.000 | 0.626 | 0.497 | 0.406 |
> > | **Nash-EGPO** | 0.583 | 0.556 | 0.496 | 0.283 | 0.000 | 0.355 | 0.263 |
> > | **Stackelberg-PG-Leader** | 0.734 | 0.613 | 0.503 | 0.633 | 0.000 | 0.395 |
> > | **Stackelberg-PG-Follower** | 0.800 | 0.656 | 0.594 | 0.722 | 0.605 | 0.000 |
> >
> > | Policy | Qwen2.5-0.5B | Qwen2.5-1.5B | RLOO | Nash-MD | Nash-EGPO| Stackelberg-PG-Leader | Stackelberg-PG-Follower |
> > | :--- | :--- | :--- | :--- | :--- | :--- | :--- | :--- |
> > | **Qwen2.5-0.5B** | 0.000 | 0.479 | 0.379 | 0.188 | 0.417 | 0.271 | 0.166 |
> > | **Qwen2.5-1.5B** | 0.521 | 0.000 | 0.401 | 0.209 | 0.444 | 0.293 | 0.187 |
> > | **RLOO** | 0.621 | 0.599 | 0.000 | 0.197 | 0.504 | 0.310 | 0.175 |
> > | **Nash-MD** | 0.812 | 0.791 | 0.803 | 0.000 | 0.717 | 0.623 | 0.489 |
> > | **Nash-EGPO** | 0.520 | 0.504 | 0.430 | 0.244 | 0.000 | 0.321 | 0.232 |
> > | **Stackelberg-PG-Leader** | 0.729 | 0.707 | 0.690 | 0.377 | 0.645 | 0.000 | 0.313 |
> > | **Stackelberg-PG-Follower** | 0.834 | 0.813 | 0.825 | 0.511 | 0.737 | 0.687 | 0.000 |
> >
> > 6. **Intransitivity Analysis**
> > Thank you for this suggestion. Our additional analysis shows nearly 19 million cycles in the dataset subset. We found the following cycle length distribution: 5.79% (length $\leq 9$), 41.64% (length 10-12), 42.72% (length 13-14), and 9.85% (length $\geq 15$). Furthermore, 33.31% of prompts have 1-9 cycles, 17.12% have 10-99, 3.63% have 100-999, and 2.54% have $\geq 1000$. We have added these detailed statistics to Appendix D.1 (revisions highlighted in color). Unfortunately, comparison to common open datasets is not feasible as they provide only a single comparison between two samples for each prompt, therefore, no cycles can be observed.
> >
> > We hope that our response, the additional experiments, and the revisions to the submission have addressed your concerns and questions. If any questions remain, we are happy to discuss further.

---

> > > ### Comment · Reviewer_Uhos · 2025-11-27
> > >
> > > Thank you for your response! I'll raise my score.

---

> > ### Comment · Reviewer_Uhos · 2025-11-27
> >
> > Regarding **Algorithm Presentation Issues**, note that in Alg 1 you performed $\pi_{i+1} = \textup{Proj} (\pi_i + \eta \nabla_\pi f(\pi_i))$, while in Alg 2 the update is $\theta_{i+1}=\theta_i - \eta \nabla_\theta L (\theta)$, these are two distinct regimes. The first one is constrained optimization while the second one is unconstrained. This difference is similar to OGDA vs OMWU. If you want to analyze the convergence guarantees, these two algorithms should use different approaches and will lead to different conditions for convergence. You can take a look at https://arxiv.org/abs/2006.09517

---

### Official Review · Reviewer_qE41 · 2025-11-01

**Soundness:** 4
**Presentation:** 4
**Contribution:** 4
**Rating:** 10
**Confidence:** 5

**Summary:**

The paper introduces Stackelberg Learning from Human Feedback (SLHF), a new framework that models alignment as a sequential two-player game between a Leader (the action proposer) and a Follower (the conditional refiner). Unlike RLHF (scalar rewards) or NLHF (simultaneous equilibria), SLHF exploits sequential asymmetry to capture richer preference structures and support inference-time refinement. The authors propose STACKELBERG-GDA to efficiently approximate equilibria and scale training to large LLMs (0.5B–8B). Empirical results show SLHF achieves strong alignment across diverse datasets, with Follower policies improving outputs even when transferred to unseen models.

**Strengths:**

This paper introduces an innovative game-theoretic Stackelberg structure for preference learning.

The proposal is rooted in the existence of intransitivity in pairwise preferences. It proposes a rational computational solution that replicates the logic with additional transparency into the learning and inference process.

Experiments showed that it outperforms or matches RLHF/NLHF baselines across multiple datasets.

Some theoretical foundations are discussed, i.e., qualitative connections to RLHF and NLHF, constructive conditions for numerical approximation, standard regularity assumptions, an equilibrium analysis, and an optimization algorithm (Stackelberg-GDA).

Source code and assets are open.

**Weaknesses:**

The two-policy framework, i.e., Leader policy and Follower policy, increases computational and training costs.

**Questions:**

Despite the discussion of model non-transitivity in Appendix D1, can you elaborate further on the merits of intransitivity coverage, compared to real-valued reward models?

In the majority of RLHF literature, people rely on 'transitivity' assumptions for its simplicity, while in real-world datasets, binary reward models, e.g., the Bradley-Terry (BT) model, when explicitly used, are known to be subject to 'intransitivity' because they rely on scalar variables that assume all preferences are transitive.

For your information, the literature below studied representative preference datasets in the real world, where the 'transitive' relationship between preference annotations may not always hold.
- https://arxiv.org/abs/2409.19325 (Duan et al., 2017)

---

> ### Author Response · Authors · 2025-11-17
>
> Dear Reviewer qE41,
>
> We would like to thank you for your time, your thoughtful review, and your supportive feedback. We are glad that you found our SLHF framework to be a sound and innovative contribution.
>
> We address your questions below.
> 1. **On higher computational and training costs due to the two-policy framework.**
> We appreciate you raising the concern of computational cost. While SLHF in principle involves two policies, we show that we can reduce training costs effectively by parameter sharing and formulating the Follower’s policy as a multi-turn dialogue as depicted in Figure 1b. This makes our memory footprint comparable to a single model. We have extended Appendix B to clarify this point (revisions highlighted in color), showing that with this practical implementation, SLHF's training costs are on par with common RLHF and NLHF baselines.
>
> 2. **Merits of intransitivity coverage**
> Real-valued reward models used in standard RLHF inherently impose transitivity, as they assume that all preferences can be represented by a single scalar score. However, as noted in prior work (Duan et al., 2017; Alós-Ferrer et al., 2022; Casper et al., 2023), real human preference data often contains cycles or local inconsistencies, and forcing these onto a scalar reward can distort the underlying structure and create sensitivity to data sampling (Munos et al., 2024), as we highlight in Section 4.1. The key merit of intransitivity coverage is that scalar rewards must arbitrarily break preference cycles, whereas pairwise methods can retain the original preference structure.
> Both NLHF and our SLHF formulation avoid assuming a transitive preference structure. In SLHF, the sequential game preserves the observed pairwise structure, including intransitivities, without flattening it into a single ordering. This allows the optimization to better reflect the structure present in the preferences.
> We appreciate the pointer to Duan et al. (2017); it has been added to our discussion of the limitations of scalar reward modeling.

---

> > ### Comment · Reviewer_qE41 · 2025-11-23
> >
> > I thank the authors for highlighting the implementation strategy and extending the discussion on intransitivity coverage. I maintain my score.

---

### Official Review · Reviewer_U2fY · 2025-11-01

**Soundness:** 3
**Presentation:** 3
**Contribution:** 3
**Rating:** 6
**Confidence:** 3

**Summary:**

This paper introduces Stackelberg Learning from Human Feedback (SLHF), a new framework for aligning LLMs that models the problem as a sequential-move game between two policies: a Leader that commits to an initial response and a Follower that refines it. This sequential approach avoids the need for a single scalar reward model, unlike traditional RLHF, allowing it to better handle complex or intransitive preferences. The paper also proposes an algorithm, STACKELBERGGDA, to find the game's solution. A key advantage of this framework is its natural ability to perform inference-time refinement, where the Follower can be used to iteratively improve the Leader's output. Experiments demonstrate that the SLHF Follower policy not only improves upon its own Leader's outputs but also consistently refines and enhances the responses from other, independently trained models without any additional fine-tuning.

**Strengths:**

- Unlike standard RLHF, SLHF optimizes directly over pairwise preferences without collapsing them into a single scalar reward, allowing it to handle complex and intransitive preference cycles.
- The Leader-Follower structure naturally supports improving model outputs at inference time, as the Follower is explicitly trained to refine a given response, allowing for iterative improvement with more computation.
- By decomposing the problem, the Follower solves a simpler refinement task against a fixed action rather than a non-stationary opponent, leading to more stable learning.

**Weaknesses:**

- The method's success heavily relies on having a "well-specified and representative pairwise preference function, which can be unavailable.
- The experiments suggest the method can be sensitive to biases in the preference judge (in this case, an "LLM-as-a-judge"). The authors attribute the gap between standard and length-controlled win rates to the judge model's "length bias," which the SLHF model may have learned to exploit.

**Questions:**

- In the practical implementation, the Leader and Follower share parameters. Could this limit the follower's ability?
- In Appendix D.3, this paper mentions that "increasing κ leads to a gradual decline in the Leader’s performance. While
the Follower benefits from increasing κ from 1 to 5". How to balance the performance of the leader and the follower? Which is more important in practice?
- Regarding "refining outputs from other models", Does it imply the Follower learns a universal refinement rather than just a policy specific to its own Leader?

---

> ### Author Response · Authors · 2025-11-17
>
> Dear Reviewer U2fY,
>
> Thank you for your time and effort in reviewing our submission. We are encouraged that you found our approach sound and appreciated its capabilities for optimising complex preferences, inference-time refinement, and improved learning stability.
> We would like to address your questions below.
>
> 1. **On Preference Functions and Judge Biases.**
> We fully agree that having a well-specified and representative preference function is crucial. However, this is a necessary for all online preference optimization algorithms (including both NLHF and our SLHF framework), and *not a limitation specific to our method*. RLHF similarly relies on the assumption that the reward function estimated with the Bradley-Terry model in Equation (2) is representative of the preferences.
> While mitigating preference model bias is a critical and inherent challenge for the entire field, our paper's primary focus is on the subsequent policy optimization step. We fully agree that addressing biases in the preference judge that we are optimizing against is an important direction in this research area.
>
> Questions:
> 1. **In the practical implementation, the Leader and Follower share parameters. Could this limit the follower's ability?**
> In preliminary experiments, we compared separate models versus a shared-parameter setup at the 0.5B scale. The separate-model version showed less stable optimization, sometimes exhibiting mode collapse, and it required roughly twice the memory footprint. The shared-weight version trained more stably while still learning clearly distinct behaviors for the Leader and Follower. It appears that weight sharing acts as a helpful regularizer. Given these observations, we adopted the shared-parameter design for all experiments. We decided to not run extensive separate-parameter variants at larger scales as it requires substantially more compute.
>
> 2. **In Appendix D.3, this paper mentions that "increasing κ leads to a gradual decline in the Leader’s performance. While the Follower benefits from increasing κ from 1 to 5". How to balance the performance of the leader and the follower? Which is more important in practice?**
> This balance is application-dependent and hinges on the inference-time compute budget. The choice of $\kappa$ presents a clear trade-off:
> * Low $\kappa$ (e.g., $\kappa=1$) prioritizes the Leader's performance. This is desirable for applications where low latency and cost are crucial (e.g., standard text completion).
> * High $\kappa$ (e.g., $\kappa=5$) improves the Follower’s refinement performance. This is preferable when maximal quality is the goal and a slight increase in latency and an additional inference step is acceptable (e.g., complex writing tasks or detailed code review).
>
> 3. **Regarding "refining outputs from other models", Does it imply the Follower learns a universal refinement rather than just a policy specific to its own Leader?**
> That is correct. Our experiments in Table 4 indicate that the Follower can refine outputs from other independently trained models as well as its own Leader. This suggests that the refinement behavior learned under SLHF is not tied to a specific Leader policy. We view this as an interesting and promising property of the SLHF formulation.
>
> We hope that this clarifies your questions. Please let us know if anything remains unclear.

---

> > ### Comment · Reviewer_U2fY · 2025-11-28
> >
> > Thanks for your response. I will maintain my score.

---

### Official Review · Reviewer_oDxv · 2025-11-02

**Soundness:** 3
**Presentation:** 3
**Contribution:** 3
**Rating:** 6
**Confidence:** 3

**Summary:**

This paper proposes a stackleberg game formulation of the RLHF problem, in contrast to prior works which either use a bradley-terry model or search for a nash equilibrium.  The authors demonstrate that their stackleberg formulation is able to resolve common problems with BT-based models, i.e. cyclical preferences, while also allowing for further test-time adaptation using the follower model.

**Strengths:**

* The paper is generally easy to follow and can be understood. The overview of differetn approaches is nice too!
* The section demonstrating the different types of preference relationships that different models can address is very nice and useful!
* The approach is well justified theoretically.
* The experimental evaluation considers both preference dataset evaluation and general finetuning
* to my knowledge, using a stackleberg game for learning from feedback is novel.

**Weaknesses:**

* The experiment section lacks any ablations on the choices made. For example, how does the two-timescale schedule affect performance?
* The method seems like it will be computationally more expensive.
* I am not sure why the stackleberg formulation makes sense. I can see how the nash formulation can resolve ambiguities in preferences vs BT, but realistically when would I want to have a leader and follower? Using the follower will double inference costs.
* the gains of the leader vs the nash models seem marginal at best. This seems to indicate that a lot of the performance gains might be coming from just using more compute / tokens for a response i.e. adding context.
* The length bias seems really strong in the Alpaca results.


Nit:
* In eq 5, the order might be more intuitive if the leader is the inner optimization and the then follower moves after? the notation for eq is also not super well defined -- and it would be nicer if the symbols for the leader and follower were more clearly introduced.

**Questions:**

* How is the reference for the follower defined? Does this reference model even make sense if the prior model hasn't been trained as a follower?
* For a lot of LLM methods, compute matters. Could the authors comment on any difference in compute requirement vs nash vs BT model + PPO? What do results look like at compute parity?
* Baselines: could the authors comment on why a method like SPO was not relevant / considered as abaseline? It is also based on nash equilibrium?

---

> ### Author Response · Authors · 2025-11-17
>
> Dear Reviewer oDxv,
>
> Thank you for your time and effort in reviewing our submission and your helpful comments. We are pleased to read that you found the paper well-structured, the approach theoretically well-justified and novel, and the experimental results comprehensive.
>
> We respond to your questions and concerns below.
>
> 1. Experiment Ablations: We would like to kindly refer you to Appendix D.3, where we already provide ablation results for the two-timescale coefficient $\kappa$. Our results show that increasing $\kappa$ initially benefits the Follower’s performance at the cost of the Leader’s while for large values it has detrimental effects for both. We also provide further results on scaling the model size in Appendix D.4. We highlight these results in the main text at the beginning of the experiments section.
>
> 2. "Why does the Stackelberg formulation make sense?” (3rd & 4th point):
> SLHF is useful because the Follower conditions on the *realized* Leader output during training, not just its distribution. This allows the Follower to learn a genuine refinement policy. The simultaneous-move Nash formulation cannot express this conditional refinement behavior, since neither player observes the other’s actual action. This is the core reason why the sequential formulation is meaningful.
>
> 3. “Wouldn’t this double inference cost, and when would I ever want to use a Follower?”: Using the Follower is optional. The Leader alone already matches the Nash baseline in **Table 3**, so SLHF does *not* require doubling inference cost to reach competitive performance. The refinement step is only used in settings where additional compute is intentionally spent to improve.
>
> 4. “Are the gains simply due to more tokens or added context?”: Our experiments suggest not. All methods receive the same additional context when evaluated as Followers in **Table 4**. If improvement came from “just more tokens”, RLOO and Nash-MD-PG would also improve when placed in the same evaluation setup. Instead, they generally degrade, while the SLHF-trained Follower improves consistently. This suggests that the relative improvement is due to the learned conditional refinement policy, not from additional tokens.
>
> 5. Regarding Eq. (5), we believe that the max min statement is in fact more intuitive, as the Leader has to move first and therefore anticipate the Follower’s min. It is also commonly written this way in the game theory literature.
>
> Questions:
>
> 1. "How is the reference for the follower defined? Does this reference model even make sense if the prior model hasn't been trained as a follower?"
> In our experiments, we use the initial policy as reference policy for both the Leader and the Follower. Due to the multi-turn formulation depicted in Figure 1, models trained for multi-turn conservations can be used as the reference policy for both the Leader and the Follower. We clarified this in Section 5 (revisions highlighted in color).
>
> 2. "For a lot of LLM methods, compute matters. Could the authors comment on any difference in compute requirement vs nash vs BT model + PPO? What do results look like at compute parity?"
> We extended our discussion on the implementation details of StackelbergGDA in Appendix B with a comparison between standard algorithms in terms of memory and compute requirements in the revised submission. While reliable comparisons are difficult to obtain as the compute requirements depend on implementation details and many algorithms include trade-offs between memory and compute, StackelbergGDA’s memory and computational costs are inline with comparable online RLHF/NLHF algorithms.
>
> 3. "Baselines: could the authors comment on why a method like SPO was not relevant / considered as a baseline? It is also based on nash equilibrium?"
> Our choice for comparison algorithms in Section 6.1 was based on three factors, which we have now clarified in Section 6:
> * Framework comparison: Our primary goal was to compare the solution concepts (RLHF vs. NLHF vs. SLHF), not algorithm runtime comparison.
> * Conceptual similarities: We chose RLOO and Nash-MD-PG as representative algorithms for RLHF and NLHF as they are also building on policy gradient updates similarly to StackelbergGDA.
> * Implementation Parity: The TRL library includes well-tested and reliable implementations of the chosen algorithms. By building StackelbergGDA similarly within TRL, we minimize performance differences due to implementation-level "tricks" and ensure a fairer comparison of the underlying algorithms. In particular, we did not consider SPO as, to the best of our knowledge, open-source implementation is not available and performance for large-scale LLM fine-tuning is not yet demonstrated.
> 	For completeness, we ran experiments with EGPO (Zhou et al. 2025) that do not show improvement compared to Nash-MD-PG. More details are provided in the rebuttal for Reviewer Uhos.
>
> We hope our answers clarify your questions. We’d be happy to elaborate further if needed.

---

> > ### Comment · Reviewer_oDxv · 2025-11-27
> >
> > Thank you for the detailed response! There are a few weaknesses I think that are worth pointing out:
> > * the length bias in alpaca eval is still really strong
> > * when given just the leader, the gains vs Nash are relatively small.
> >
> > However, these are largely experimental and I think the paper has some very interesting ideas, especially when considering further refinement. I have updated my score accordingly.

---

### Meta-Review · Area_Chair_QRKj · 2025-12-29

**Summary:**

In this paper, the authors propose a Stackleberg game formulation of RLHF and show that their formulation can address some of the major problems of the Bradley-Terry-based models, such as cyclical preferences, while also allowing for test-time adaptation using the follower model.

All reviewers found the paper well-motivated, well-written, and easy-to-follow. They also found the proposed formulation novel and well-supported by theoretical justifications and empirical evidence. The authors successfully show that their proposed formulation is capable of addressing some of the shortcomings of the existing methods, performs as well or better than them, and finally the leader-follower structure proposed in the paper can naturally support inference-time improvement, which is important in LLM applications.

I would like to ask the authors to take the reviewers' comments into account in preparing the final version of the paper.

**Reviewer Concerns:**

The authors successfully addressed most of the reviewers' concerns.

**Reviewer Scores:**

Reviewers U2fY and qE41 decided to maintain their scores (6 and 10, respectively) and Reviewers oDxv and Uhos decided to raise their scores, probably from 6 to 8 and from 4 to 6, respectively.

---

### Decision · Program_Chairs · 2026-01-26

Accept (Poster)